# Adversarially-Trained Deep Nets Transfer Better: Illustration on Image Classification

**Francisco Utrera**[*]
UC Berkeley
utrerf@berkeley.edu

**Evan Kravitz**[*]
UC Berkeley
kravitz@berkeley.edu

**N. Benjamin Erichson**
ICSI and UC Berkeley
erichson@berkeley.edu

**Rajiv Khanna**
UC Berkeley
rajivak@berkeley.edu

**Michael W. Mahoney**
ICSI and UC Berkeley
mmahoney@stat.berkeley.edu

## Abstract

Transfer learning has emerged as a powerful methodology for adapting pre-trained deep neural networks on image recognition tasks to new domains. This process consists of taking a neural network pre-trained on a large feature-rich source dataset, freezing the early layers that encode essential generic image properties, and then fine-tuning the last few layers in order to capture specific information related to the target situation. This approach is particularly useful when only limited or weakly labeled data are available for the new task. In this work, we demonstrate that adversarially-trained models transfer better than non-adversarially-trained models, especially if only limited data are available for the new domain task. Further, we observe that adversarial training biases the learnt representations to retaining shapes, as opposed to textures, which impacts the transferability of the source models. Finally, through the lens of influence functions, we discover that transferred adversarially-trained models contain more human-identifiable semantic information, which explains – at least partly – why adversarially-trained models transfer better.

## 1 Introduction

While deep neural networks (DNNs) achieve state-of-the-art performance in many fields, they are known to require large quantities of reasonably high-quality labeled data, which can often be expensive to obtain. As such, transfer learning has emerged as a powerful methodology that can significantly ease this burden by enabling the user to adapt a pre-trained DNN to a range of new situations and domains (Bengio, 2012; Yosinski et al., 2014). Models that are pre-trained on ImageNet (Deng et al., 2009) have excellent transfer learning capabilities after fine-tuning only a few of the last layers (Kornblith et al., 2019) on the target domain.

Early work in transfer learning was motivated by the observation that humans apply previously learned knowledge to solve new problems with ease (Caruana, 1995). With this motivation, learning aims to extract knowledge from one or more source tasks and apply the knowledge to a target task (Pan & Yang, 2009). The main benefits include a reduction in the number of required labeled data points in the target domain (Gong et al., 2012; Pan & Yang, 2009) and a reduction in training costs as compared to training a model from scratch. However, in practice, transfer learning remains an "art" that requires domain expertise to tune the many knobs of the transfer process. An important consideration, for example, is which concepts or features are transferable from the source domain to the target domain. The features which are unique to a domain cannot be transferred, and so an important goal of transfer learning is to hunt for features shared across domains.

It has recently been shown that adversarially-trained models (henceforth denoted as **robust** models) capture more *robust* features that are more aligned with human perception, compared to the seemingly patternless features (to humans, at least) of standard models (Ilyas et al., 2019). Unfortunately,

---

[*]Equal contribution

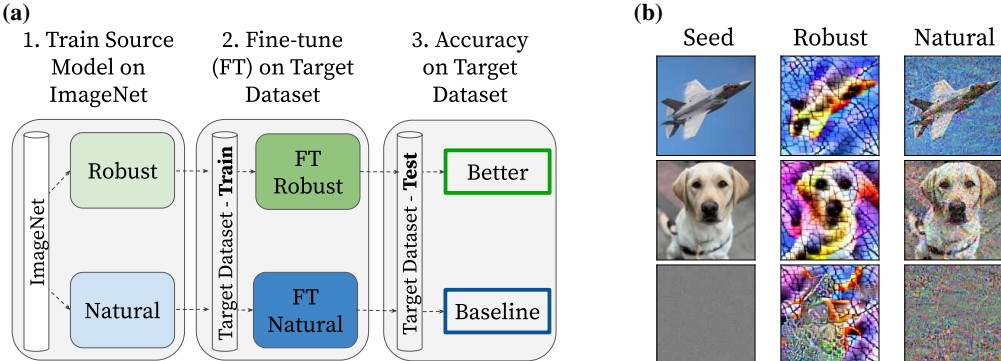

Figure 1: We demonstrate that adversarially-trained (i.e., robust) DNNs transfer better and faster to new domains with the process shown in (a): A ResNet50 is trained adversarially or non-adversarially (i.e., naturally) on the source dataset. Then, we fine-tune both of these source models on the target dataset. We hypothesize the *robust* features in robust models that encode more humanly perceptible representations, such as textures, strokes and lines, as seen in (b), are responsible for this phenomenon. See Appendix A.1 for details on how we generated the images in (b).

these models typically have a *lower* generalization performance on the source domain, as compared to non-adversarially-trained (henceforth denoted as **natural**, as in previous works (Tsipras et al., 2019; Shafahi et al., 2019; Salman et al., 2020)) model. Hence, Ilyas et al. (2019) hypothesize that *non-robust* features that are lost during adversarially training may have a significant positive impact on generalization *within* a given dataset or domain. This inherently different feature representation between models constructed with adversarial training and models trained with standard methods would also explain why accuracy and robustness are at odds (Tsipras et al., 2019). This leads to the question of whether models that use robust representations generalize better *across* domains. This is the main question we address.

In this work, we demonstrate that robust models transfer better to new domains than natural models. To demonstrate this, we conduct an extensive number of transfer learning experiments across multiple domains (i.e., datasets), with various numbers of fine-tuned convolutional blocks and random subset sizes from the target dataset, where the critical variable is the constraint used to adversarially train the source model. (Described in detail in Sections 3 and Appendix A.3) Importantly, note that we do not use an adversarial training procedure for the actual transfer learning process. Our findings indicate that robust models have outstanding transfer learning characteristics across all configurations, where we measure the performance in terms of model accuracy on target datasets for varying numbers of training images and epochs. Figure 1 provides a summary of our approach.

Our focus in this work is to show that robust source models learn representations that transfer better to new datasets on image recognition tasks. While adversarial training was proposed to combat adversarial attacks, our experiments discover an unintended but useful application. Adversarial training retains the robust features that are independent of the idiosyncrasies present in the source training data. Thus, these models exhibit worse generalization performance on the source domain, but better performance when transferred. This observation is novel, and we undertake extensive empirical studies to make the following contributions:

- We discover that adversarially-trained source models obtain higher test accuracy than natural source models after fine-tuning with fewer training examples on the target datasets and over fewer training epochs.

- We notice that the similarity between the source and target datasets affects the optimal number of fine-tuned blocks and the robustness constraint.

- We show that adversarial training biases the learned representations to retain shapes instead of textures, impacting the source models' transferability.

- We interpret robust representations using influence functions and observe that adversarially-trained source models better capture class-level semantic properties of the images, consistent with human concept learning and understanding.

## 2 RELATED WORKS

**ImageNet transfers.** Our focus is on studying the transfer of all but the last few layers of trained DNNs and fine-tuning the last non-transferred layers. For ease of exposition, we restrict our attention to ImageNet models (Deng et al., 2009). Kornblith et al. (2019) study the transfer of natural models to various datasets and is thus a prequel to our work. Yosinski et al. (2014) also study transferring natural models but focus on the importance of individual neurons on transfer learning. Recht et al. (2019) study the generalization of natural and robust models to additional data generated using a process similar to that of generating ImageNet. They conclude that models trained on ImageNet overfit the data. However, they study the models' generalization as-is without fine-tuning.

**Covariate shift.** A significant challenge in transfer learning is handling the data distribution change across different domains, also called covariate shift. It's widely recognized in successful domain adaptations (Yosinski et al., 2014; Glorot et al., 2011) that the representations in earlier layers are more "generic" and hence more transferable than the ones in later layers. This hierarchical disentanglement is attributed to the properties of the data itself, so that the later layers are more closely associated with the data and do not transfer as well. This motivated studies for shallow transfer learning (Yosinski et al., 2014; Ghifary et al., 2014) and more general studies to extract features that remain invariant across different data distributions (Arjovsky et al., 2019). In Section 5 we see that adversarial training biases the learned representations to retain shapes instead of textures, which may be a more desirable invariant across the datasets.

**Transfering adversarially-trained models.** There are mainly two works directly associated with ours. First, subsequent to this paper's initial posting (in a non-anonymized form in a public forum), Salman et al. (2020) posted a related paper. They arrived at broadly similar conclusions, confirming our main results that robust models transfer better; and they do so by focusing on somewhat different experiments, e.g., they focus on the effects of network architecture width, fixed feature transfer, and seeing if models without texture bias transfer better than robust models. Second, Shafahi et al. (2020) mainly find that models lose robustness as more layers are fine-tuned. It might seem to contradict our thesis that they also notice that an ImageNet robust model with a $\|\delta\|_\infty \leq 5$ constraint has lower accuracy on the target datasets, CIFAR-10 and CIFAR-100, compared to a natural ImageNet model. However, we show that the robust model transfers better than the natural one when we use a $\|\delta\|_2 \leq 3$ constraint to adversarially train the source model.

**Example based interpretability.** There has been significant interest in interpreting blackbox models using salient examples from the data. A line of research focuses on using influence functions (Koh & Liang, 2017; Koh et al., 2019; Khanna et al., 2019) to choose the most indicative data points for a given prediction. In particular, Khanna et al. (2019) discuss the connection of influence functions with Fisher kernels; and Kim et al. (2016) propose using criticisms in addition to representative examples. Complimentary lines of research focus on interpretability based on human-understandable concepts (Bau et al., 2017) and feature saliency metrics (M. Ancona, 2017).

## 3 BRIEF OVERVIEW OF THE ADVERSARIAL TRAINING PROCESS

Adversarial training modifies the objective of minimizing the average loss across all data points by first maximizing the loss produced by each image with a perturbation (i.e., a mask) that may not exceed a specified magnitude. Here, we describe this process, similar to Madry et al. (2018).

Let $(x_i, y_i)$ be $m$ data points for $i \in [m]$, where $x_i \in \mathbb{R}^d$ is the $i^{\text{th}}$ feature vector, and $y_i \in \mathcal{Y}$ is the corresponding response value. Typically, we model the response as a parametric model $h_\theta : \mathbb{R}^d \to \mathcal{Y}$ with a corresponding loss function $\ell : \mathcal{Y} \times \mathcal{Y} \to \mathbb{R}_{\geq 0}$. The objective is to minimize the loss $\ell(\hat{y}, y)$, where $\hat{y} = h_\theta(x)$ is the predicted response. Adversarial training replaces the above minimization problem of training the model by a minimax optimization problem to make the model resilient to arbitrary perturbations of inputs. The goal of *adversarial training* is to solve a problem of the form

$$\min_\theta \frac{1}{m} \sum_{i=1}^m \max_{\|\delta_i\|_p \leq \epsilon} \ell(h_\theta(x_i + \delta_i), y_i). \tag{1}$$

That is, the goal is to find the parameters $\theta$ of the model $h_\theta$ that minimize the average maximum loss obtained by perturbing every input $x_i$ with a $\delta_i$ constrained such that its $\ell_p$ norm does not exceed

some non-negative $\epsilon$. If $\epsilon = 0$, then $\delta_i = \mathbf{0}$, in which case there is no perturbation to the input, which is what we call *natural training*. As $\epsilon$ increases, the magnitude of the perturbation also increases. For more details on how we solve this problem, and a few examples, see Appendix A.2.

## 4 TRANSFERRING ADVERSARIALLY-TRAINED MODELS

In this study, we train four ResNet50 source models on ImageNet. We train one of them naturally (non-adversarially), and train each of the remaining three adversarially with one of the following constraints: (i) $\|\delta\|_2 \le 3$, (ii) $\|\delta\|_\infty \le \frac{4}{255}$, (iii) $\|\delta\|_\infty \le \frac{8}{255}$. Next, we fine-tune some convolutional blocks in the source models to each of the six target datasets separately using a subset of the training data. We repeat each of these trials for various seed values and report the mean and 95% confidence interval. Altogether, we have a comprehensive and replicable experimental setup that considers four ImageNet source models, four fine-tuning configurations, six target datasets, ten random subset sizes, and an average of fifteen random seeds for a total of 14,400 fine-tuned models. For more details, see Appendix A.3 and A.4.

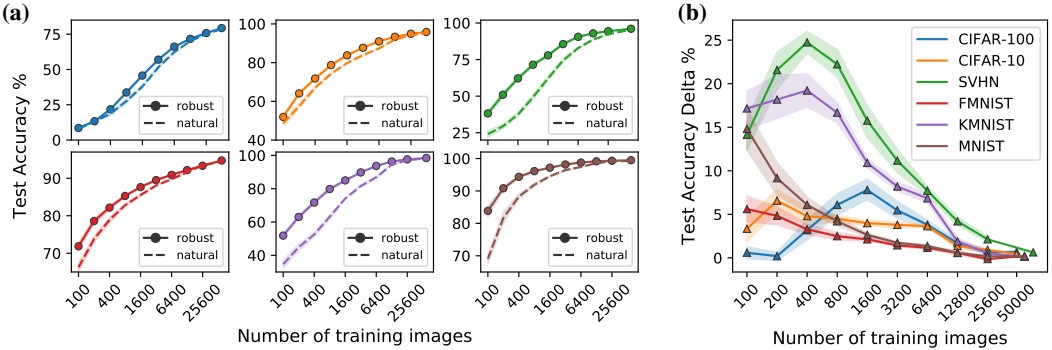

Figure 2: Robust models generalize better to new domains, especially with fewer training images in the target domain. (a) Shows the test accuracy on the six target datasets (color-coded as in (b)) for various subset sizes. (b) Shows the test accuracy delta, defined as the robust model test accuracy minus natural model test accuracy. The solid line is the mean and its shade is the 95% confidence interval. Both the robust and natural models are ResNet50s' trained on ImageNet. The robust model uses a $\|\delta\|_2 \le 3$ constraint. Both models fine-tune three convolutional blocks on the target dataset.

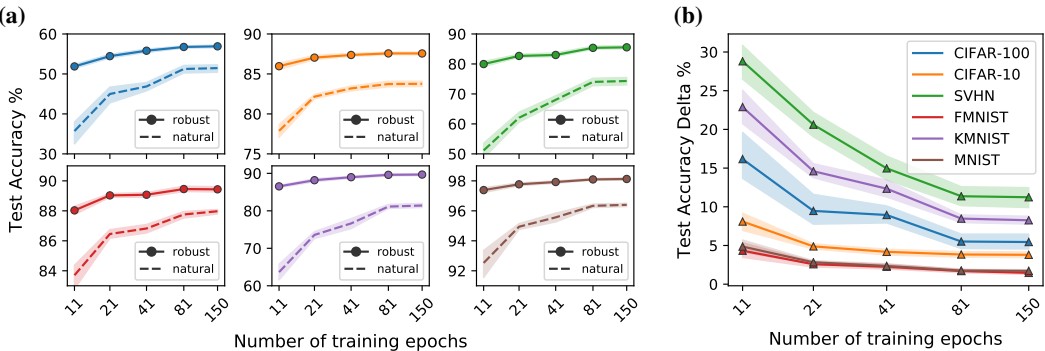

Figure 3: Robust models transfer faster. (a) Shows the test accuracy during the fine-tuning process on the six target datasets (color-coded as in (b)). (b) Shows the test accuracy delta, defined as the robust model test accuracy minus the natural model test accuracy. The solid line is the mean and its shade is the 95% confidence interval. Both the robust and natural models are ResNet50s' trained on ImageNet. The robust model uses a $\|\delta\|_2 \le 3$ constraint. Both models fine-tune three convolutional blocks using a random subset of 3,200 images ($\sim 5\%$) of the target dataset.

**Adversarially-trained models transfer better and faster.** For ease of comparison, we select the robust and natural models that transfer with the highest test accuracy across all datasets (fine-tuning three convolutional blocks and the robust model using the $\|\delta\|_2 \leq 3$ constraint), as shown in Figures 2 and 3. See Appendix A.5 for additional results. Figure 2(b) shows that the test accuracy delta between robust and natural models is above zero for all six target datasets. Thus, robust models obtain higher test accuracy on the target dataset than the natural model, especially with less training data in the target domain. Robust models also learn faster, as shown by the positive test accuracy delta in Figure 3(b) for all target datasets after only 11 and 21 fine-tuning epochs. See Appendix A.6 for additional information on different random subset sizes. Fine-tuning cost is the same for both robust and natural models, but training the source model is considerably more expensive. For more detail on computational complexity see A.8. Also, our code is available at https://github.com/utrerf/robust_transfer_learning.git

**Best results achieved with $\ell_2$ constraint and fine-tuning one to three convolutional blocks.** Robust models achieve the highest test accuracy on the target datasets when an optimal number of convolutional blocks are fine-tuned, and when these models are trained with an appropriate constraint type. In particular, fine-tuning zero (only the fully-connected layer) or nine convolutional blocks leads to lower test accuracy than fine-tuning one or three blocks, as shown in Figure 4(a) for all six target datasets. The natural model and the other two robust models exhibit the same behavior, as shown in Appendix A.7. To analyze the best constraint type, we select the fine-tuning configuration that yields the highest test accuracy on the target datasets (fine-tuning three convolutional blocks). We see that the $\ell_2$ constraint outperforms the $\ell_\infty$ constraint, as shown by the positive accuracy delta between the $\ell_2$ and $\ell_\infty$ models in Figures 5(d) and (e), respectively.

**Similarity effect on transfer learning configurations.** Besides noticing that robust models achieved better performance on the target dataset than natural models, we also observe trends in how well they transfer to different datasets. When transferring from ImageNet, we find that CIFAR-10 and CIFAR-100 have interesting transfer properties, compared to the other datasets. In particular, even though all other datasets transfer better when fine-tuning one or three blocks, it seems that models transfer better to CIFAR-10 and CIFAR-100 when fewer blocks are fine-tuned, as shown in Figure 4(b). This suggests that because these datasets are close to ImageNet, fine-tuning of early blocks is unnecessary (Yosinski et al., 2014). Along similar lines, it is better to use a smaller $\epsilon$ for CIFAR-10 and CIFAR-100 datasets than the other datasets when transferring from ImageNet, as seen from Figure 5(c). This is because a larger perturbation would destroy low-level features, learned from ImageNet, which are useful to discriminate between labels in CIFAR-10 and CIFAR-100. Finally, for datasets that are most distinct from ImageNet (SVHN and KMNIST), we find that robustness yields the largest benefit to classification accuracy and learning speed, as seen in Figure 2(b) and Figure 3(b), respectively. These discrepancies are even more noticeable when smaller fractions of the target dataset are used.

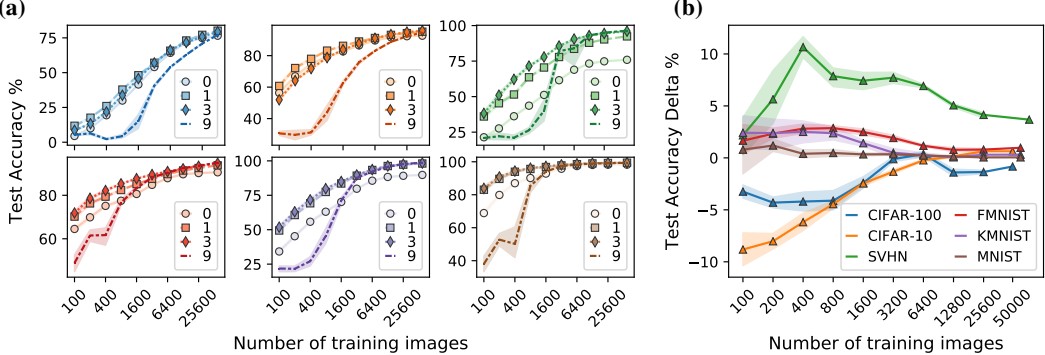

Figure 4: The optimal number of fine-tuned blocks is somewhere between one and three. (a, b) Shows the test accuracy of the robust model trained on ImageNet using the $\|\delta\|_2 \leq 3$ constraint with various numbers of fine-tuned blocks (0, 1, 3, or 9). (a) Shows the test accuracy on each of the six target datasets (color-coded as in (b)). (b) Shows the test accuracy delta, defined as the test accuracy of the model with three fine-tuned blocks minus the test accuracy of the model with one fine-tuned block. The solid line is the mean and its shade is the 95% confidence interval.

## 5 BIAS TOWARDS RECOGNIZING SHAPES AS OPPOSED TO TEXTURES

In this section, we explore the effect of texture and shape bias, as described by Geirhos et al. (2019), on the robust models' transferability. As pointed out by Geirhos et al. (2019), natural models are more biased towards recognizing textures than shapes. This is in stark contrast to the human bias of recognizing shapes over textures (Landau et al., 1988). However, Engstrom et al. (2019) showed that robust models encode humanly-aligned representations, and we observe (e.g see Figure 1(b)) that these representations persist even after fine-tuning on CIFAR-10.

**Adversarially-trained models are less sensitive to texture variations.** Table 1a shows that the robust model outperforms the natural one when only tested on Stylized Imagenet (SIN) and also after fine-tuning only the last fully-connected layer to SIN. Both models are ResNet50s pre-trained on ImageNet (IN), and the robust model uses a $\|\delta\|_2 \leq 3$ constraint.

**Models trained on standard and stylized ImageNet are less sensitive to adversarial attacks.** Table 1b shows that the ResNet50 model trained on both IN and SIN (IN+SIN) outperforms the models trained on just IN on a PGD(3) adversarial test accuracy on IN for various $\epsilon$ levels.

**Adversarially-trained models are biased towards low resolution and low frequencies.** We observe that the transferability of robust models is also affected by two input perturbations that destroy, or at least damage, textures. Namely, lowering the resolution of images and applying low pass filters. To demonstrate this, we use the Caltech101 dataset (Li Fei-Fei et al., 2004). This dataset has 101 labels with 30 high-resolution (224x224 pixels or more) images per label. The results in Table 2 support our conjecture that robust models use shapes more than textures for classification by

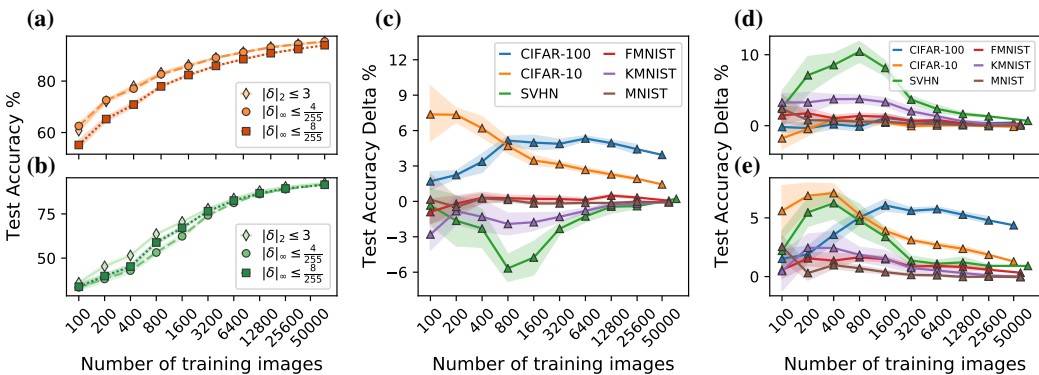

Figure 5: Shows the test accuracy on target datasets, (a) CIFAR-10 and (b) SVHN, of three robust models. (c) Shows that source models trained with lower $\epsilon$ transfer better to target domains that are similar to ImageNet, as shown by the test accuracy delta between the $\|\delta\|_\infty \leq \frac{4}{255}$ model and the $\|\delta\|_\infty \leq \frac{8}{255}$ model. (d, e) Shows the test accuracy delta between the $\ell_2$ norm with $\epsilon = 3$ and each of the two $\ell_\infty$ models: $\epsilon = \frac{4}{255}$ in (d), and $\epsilon = \frac{8}{255}$ in (e). In both (d, e) the $\ell_2$ norm constraint outperforms both of the $\ell_\infty$ constraints.

Table 1: Adversarially-trained models are less biased towards recognizing textures than natural ones. Models trained on both ImageNet and Stylized ImageNet are more robust to adversarial attacks.

(a) Shows SIN test accuracy of robust and natural IN source models before (i.e., SIN (Test)) and after fine-tuning (i.e., SIN (FT)) three convolutional blocks. Robust constraint: $\|\delta\|_2 \leq 3$.

| Model | SIN (Test) | SIN (FT) |
|---|---|---|
| Robust | **20.1** | **64.2** |
| Natural | 11.4 | 36.1 |

(b) Shows IN PGD(3) adversarial test accuracy with various $\epsilon$ magnitudes for ResNet50 models naturally-trained (non-adversarially) on IN, SIN, and both IN and SIN.

| $\epsilon$ (Test) | IN | SIN | IN+SIN |
|---|---|---|---|
| 3/32 | 52.1 | 39.6 | **53.9** |
| 3/16 | 34.7 | 25.5 | **38.5** |

showing that the robust model obtains a higher test accuracy, in both the low-resolution and low-pass versions of Caltech101, than the natural one.

# 6 INTERPRETING REPRESENTATIONS USING INFLUENCE FUNCTIONS

In this section, we use influence functions (Koh & Liang, 2017) to show that robust representations hold semantic information, i.e., robust DNNs classify images like a human would, through similar-looking examples. Engstrom et al. (2019) observed that moving the image in carefully chosen directions in the latent space allows for high-level human-understandable feature manipulation in the pixel space. They suggest that the bias introduced by adversarial training can be viewed as a *human prior* on the representations, so that these representations are extractors of high-level human-interpretable features. It has long been established that humans learn new concepts through concept-representative or similar-looking examples (Cohen et al., 1996; Newell, 1972). Our focus in the present work is to study whether these representations aid the neural network to *learn* new concepts (namely image labels) akin to how humans learn concepts.

To study this, we use influence functions as described by Koh & Liang (2017) (see Appendix A.9 for an overview). For each test image in the CIFAR-10 dataset, influence functions allow us to answer the following: What is the influence of each training image on the model prediction for a given test image? We ask this question for both the robust and natural models, and compare the results. In our experiments, we fine-tune the last three blocks with the same 3,200 randomly selected training images. Also, the robust model uses $\|\delta\|_2 \leq 3$ as the constraint.

**Adversarially-trained models have more similar-looking influential images.** Figure 6 shows that the robust models' most influential image is often more perceptibly similar to the test image than the natural models' most influential image. Consider, for example, the test image of the blue car (on the second column). The robust models' corresponding top influential training image is a similar-looking blue car, while the natural model has a red truck. As a second example, the robust

Table 2: Robust models transfer better than natural ones on low resolution and low pass filtered variants of Caltech101. This table shows the test accuracy after fine-tuning three convolutional blocks of the robust model ($\|\delta\|_2 \leq 3$) and the natural model on a low resolution (Low-Res) or a low pass filtered (Low-Pass) version of Caltech101. We lowered the resolution to 32x32 and zeroed out the top 1,024 ($\sim 14\%$) frequencies, respectively.

| Model | Low-Res | Low-Pass |
|---|---|---|
| Robust | **83.7** | **85.3** |
| Natural | 79.1 | 80.7 |

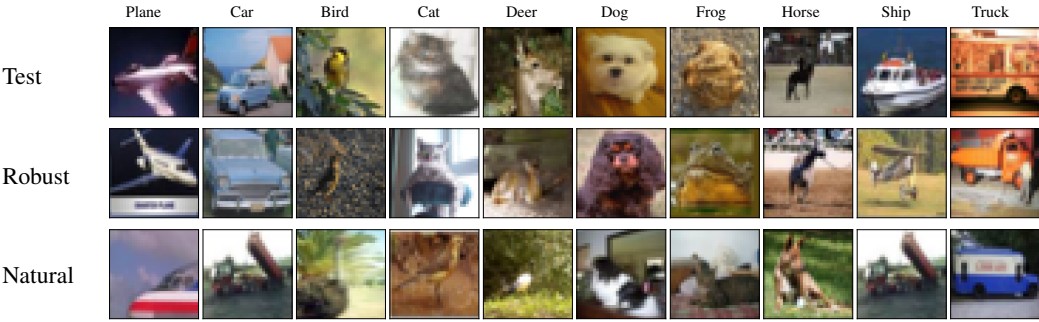

Figure 6: Adversarially-trained (i.e, robust) models have more similar-looking influential images in the target dataset than the non-adversarially-trained (i.e., natural) model. The top row shows a randomly selected test image, for each of the ten categories. The middle and bottom rows display the most influential images, for the robust and natural models, respectively.

**(a)**

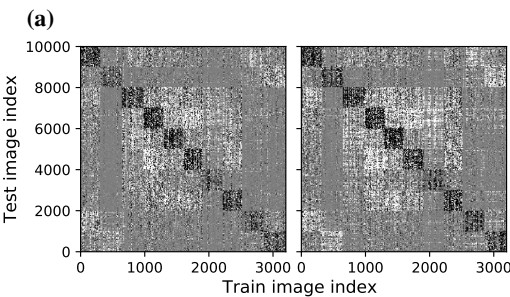
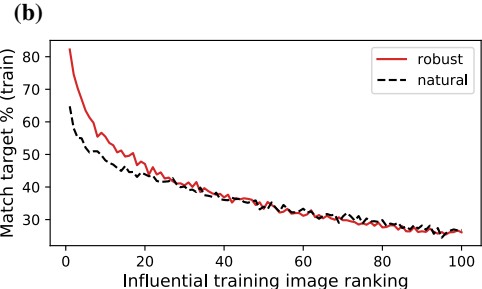

**(b)**

Figure 7: Top influential image labels in the robust model match test image labels more often than in the natural model. (a) Shows the influence values (standardized by their matrix norm) of each training image on each test image, sorted by their label as in Figure 6, for both the natural (left) and robust (right) models. (b) Displays the percentage of times that the label of the top-$k$ influential image in the training set matches the label in the test image being evaluated.

models' top influential training image for the orange truck (on the far right) is a similar-looking orange truck, while the natural model has a blue and white truck.

**Influential image labels match test image labels more often in adversarially-trained models.** To quantify the visual similarity described above, we show the influence values (standardized by their matrix norm) of each training image on each test image, sorted by their label as in Figure 6, for both the natural (left) and robust (right) models in Figure 7(a). Darker and better-defined blocks across the diagonal signal that the influence values are more consistent with the test image label index in the y-axis because darker colors represent higher influence values. The robust model (right) has a slight advantage over the natural model.

Figure 7(b) further accentuates the difference between the robust model and the natural model. It displays the percentage of times that the label of the top-$k$ influential image in the training set matches the label in the test image evaluated. To better understand this figure, consider the leftmost point in Figure 7(b) for both models. This point represents the proportion of the training images corresponding to the darkest dots in each horizontal line (i.e., top-1 influential training image) in (a) that match the label of the given test image, for robust and natural models separately. $78.6\%$ of the robust model's top-1 influential images match the label of the given test image vs $55.1\%$ for the natural counterpart. We also consider the case when the category of at least three of the top-5 influential training images match that of the test image. This happens in $77.3\%$ of the cases for the robust model, but only for $53.8\%$ of the cases for the natural model. This vast gap is not explainable solely from only $\sim 5\%$ difference in target test accuracy, shown in Table 7 in Appendix A.5.

From the qualitative and quantitative analysis, we see that the robust model has learned representations with more human-identifiable semantic information than the natural model, while the latter relies on less interpretable representations. In other words, the robust neural network has *learned* the image labels by creating strong associations to semantically-similar examples (akin to example-based concept learning in human beings) in its internal representations. Thus, reinforcing the *human prior* bias hypothesis in robust representations observed by Engstrom et al. (2019).

## 7  DO OTHER ADVERSARIAL ATTACKS IMPROVE TRANSFERRABILITY?

Prior works show that there is a connection between the sensitivity of a neural network to Gaussian noise and its robustness to adversarial perturbations (Weng et al., 2018; Gilmer et al., 2019). It has also been suggested that Gaussian perturbations can improve or even replace adversarial training (Kannan et al., 2018). Further, it has been shown that often only a few PGD iterations are sufficient to obtain a robust model (Madry et al., 2018; Shafahi et al., 2019; Wong et al., 2020).

To better understand these trade-offs, in this section we further explore the transferrability of models trained on ImageNet with random Gaussian noise and one-step of PGD (i.e., PGD(1)) using the same methodology as described in Section 4. For all models, including the Gaussian one, we contraint the

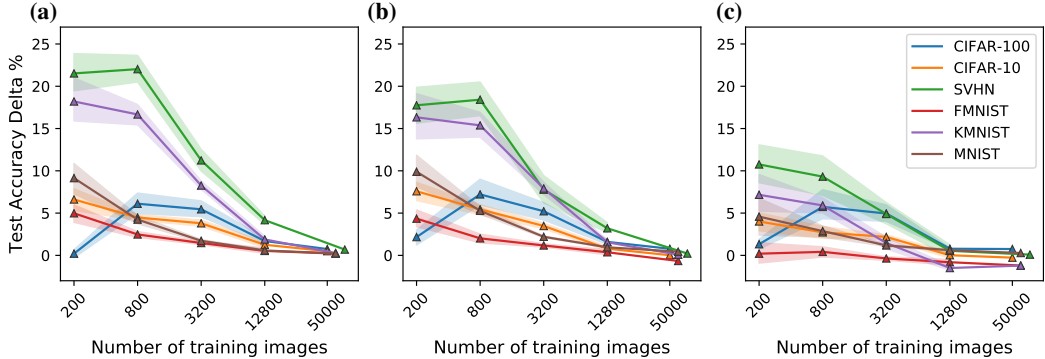

Figure 8: Shows the test accuracy delta, defined as the transfer test accuracy of the source models trained with PGD(20) in (a), PGD(1) in (b), and Gaussian in (c), minus the naturally-trained model. Comparing (a) and (b) shows that more attack steps (i.e., PGD(20) vs PGD(1)) slightly improves transferability. Comparing (a) and (b) to (c) shows that a targeted adversarial attacks (i.e. PGD vs Gaussian) significantly improves transferability relative to random ones. (c) shows us that a random adversarial attack can improve transfer accuracy.

perturbation, namely $\delta$, to be $\|\delta\|_2 \leq 3$ in order make a fair comparison across models. See A.10 for more details on our experimental setup.

In the following we discuss our experimental results, summarized in Figure 8, which shows the test accuracy delta of each of the three adversarially-trained models versus the natural one.

**Training with more steps is marginally better.** Figure 8 (a) and (b) show that more PGD iterations (i.e., PGD(20) vs PGD(1)) slightly improve transferability. This is evidenced by the slightly higher test accuracy delta in (a), relative to (b) across all target datasets. Our emperical result agrees with previous works that are showing that more attacker steps typically only improve adversarial robustness slightly (Madry et al., 2018; Shafahi et al., 2019; Wong et al., 2020).

**A targeted adversary is better than a random one.** Comparing Figure 8 (a) and (b) to (c) shows that a targeted adversarial attacks (i.e. PGD vs Gaussian) significantly improves transferability relative to a random perturbation. This is evidenced by the slightly higher test accuracy delta in (a), and (b) relative to (c) across all target datasets.

**A random adversary is better than no adversary.** Figure 8 (c) shows us that a random adversarial attack can improve transferability. This is evidenced by the significantly positive accuracy delta in (c) across all target datasets. Our results agree with prior works showing that training a model by perturbing inputs with Gaussian noise can improve adversarial robustness (Kannan et al., 2018).

# 8 CONCLUSION AND FUTURE WORKS

We show that robust models transfer very well to *new* domains, even outperforming natural models. This may be surprising since robust models generalize worse than natural models *within* the source domain, and since they were originally designed to protect against adversarial attacks. We show that robust DNNs can be transferred both faster and with higher accuracy, while also requiring fewer images to achieve suitable performance on the target domain. We observe that adversarial training biases the learned representations to retain shapes instead of textures, which impacts the source models' transferability. We also show that the improved classification accuracy is due to the fact that robust models have an implicit bias that enables them to comprehend human-aligned features. Given the widespread use of DNNs, there is great potential for robust networks to be applied to a variety of high-tech areas such as facial recognition, self-driving cars, and healthcare, but understanding the issues we have addressed is crucial to deliver upon that potential. Please see Appendix A.12 for details on future works.

ACKNOWLEDGMENTS

We are grateful to the generous support from Amazon AWS and Google Cloud. NBE and MWM would like to acknowledge IARPA (contract W911NF20C0035), NSF, ONR and CLTC for providing partial support of this work. Our conclusions do not necessarily reflect the position or the policy of our sponsors, and no official endorsement should be inferred.

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

## A    APPENDIX

### A.1    DETAILS ON FIGURE 1(A)

We generated Figure 1(a) by maximizing the activations in the penultimate layer on out of distribution images (i.e., images that are are not part of the training and test data) as our starting seeds, following the "feature visualization" technique as implemented in Engstrom et al. (2019). The natural and robust models are pre-trained ResNet-50 models on ImageNet after fine-tuning only the last fully-connected layer on CIFAR-10. The robust model used the $\|\delta\|_2 \leq 3$ constraint.

### A.2    MORE DETAILS ON ADVERSARIAL TRAINING

In practice, we solve equation 1 using stochastic gradient descent (SGD) over $\theta$. More concretely, we include a random sample of training examples in a set $B$, specify a non-negative learning rate $\alpha$, calculate the gradient with respect to the parameters of the model $\nabla_\theta$, and update $\theta$ as follows:

$$\theta := \theta - \frac{\alpha}{|B|} \sum_{(x_i, y_i) \in B} \nabla_\theta \max_{\|\delta_i\|_p \leq \epsilon} \ell(h_{\theta_i}(x_i + \delta_i), y_i). \tag{2}$$

This training process has a sequential nature: it first finds the worst possible perturbation $\delta_i^*$ for each training example $(x_i, y_i)$ in $B$ before updating the parameters of the model $\theta$, where

$$\delta_i^* = \arg\max_{\|\delta_i\|_p \leq \epsilon} \ell(h_\theta(x_i + \delta_i), y_i). \tag{3}$$

Problem equation 3 is typically solved using projected gradient descent with $k$ update steps, which is what we call PGD($k$). In this work, we use PGD(20), which means that we take 20 update steps to solve (3). Each step iteratively updates $\delta_i$ by projecting the update onto the $\ell_p$ ball of interest:

$$\delta_i := \mathcal{P}(\delta_i + \alpha \nabla_{\delta_i} \ell(h_\theta(x_i + \delta_i), y_i)). \tag{4}$$

As an example, consider the case of the $\ell_2$ norm and let $f(\delta_i) = \ell(h_\theta(x_i + \delta_i), y_i)$. If we want to meet the restriction that $\|\delta_i\|_2 \leq \epsilon$, we can pick an update value for $\delta_i$ whose $\ell_2$ norm will be at most the learning rate. This yields the problem:

$$\arg\max_{\|v\|_2 \leq \alpha} v^\top \nabla_{\delta_i} f(\delta_i) = \epsilon \frac{\nabla_{\delta_i} f(\delta_i)}{\|\nabla_{\delta_i} f(\delta_i)\|_2}. \tag{5}$$

And in the case of the $\ell_\infty$ norm we have

$$\arg\max_{\|v\|_\infty \leq \alpha} v^\top \nabla_{\delta_i} f(\delta_i) = \epsilon \cdot \text{sign}(\nabla_{\delta_i} f(\delta_i))$$

Thus, we set the learning rate to be equal to $\alpha = c \cdot \frac{\epsilon}{\text{num. of steps}}$ for $1.5 < c < 4$ in order to ensure that we reach the boundary condition for $\delta_i$. Also we must clip $\delta_i$ according to the $\ell_p$ norm in case that it exceeds the boundary condition.

### A.3    TRANSFER LEARNING EXPERIMENTAL SETUP DETAILS

**Source models.** For all of our experiments, we use four residual networks (ResNet-50) (He et al., 2016) pre-trained on the ImageNet dataset (Deng et al., 2009), one was naturally-trained (without

an adversarial constraint), and the others use PGD(20) and the following adversarial constraints: (1) $\|\delta\|_2 \leq 3$, (2) $\|\delta\|_\infty \leq \frac{4}{255}$, (3) $\|\delta\|_\infty \leq \frac{8}{255}$, where $\delta$ is a matrix that contains represents the perturbation applied to the input image as described in Section 3, equation (1). In addition to the natural ResNet-50, considered as baseline, we also use three robust networks with various constraints. For speed, transparency and reproducibility, we do not re-train the source models ourselves.[1]

**Fine-tuning procedure.** To transfer our models we copy the entire source model to the target model, freeze the all but the last $k$ convolutional blocks, re-initialize the last fully-connected (FC) layer for the appropriate number of labels, and *only* fine-tune (re-train) the last FC layer plus 0, 1, 3, or 9 convolutional blocks. Freezing layers in the neural networks entails permitting forward propagation, but disabling the back-propagation of gradients used during SGD training. We have four different fine-tuning configurations, one for each number of fine-tuned convolutional blocks. Note that the ResNet model that we consider has residual blocks that are composed of three convolutional layers, i.e., we fine-tune 27 layers plus the fully connected layer when the number of fine-tuned blocks is equal to 9. (See Section A.4 for a visualization of our fine-tuning process).

**Random subsets.** One of the most interesting parts of our experimental setup is that we also explore the test accuracy of our fine-tuned models using randomly chosen subsets of 100, 200, 400, ... , and 25,600 images from the target dataset. These subsets are constructed using random sampling without replacement, with a minor constraint: all labels must have at least one training image. For each run of model training, we fix the training data to be a randomized subset of the entire training data. As the number of images in a random subset decreases, the variance in the validation accuracy of the transferred models increases. Thus, we repeat the fine-tuning procedure using 20 seeds for every subset with at most 1,600 images, and using 10 seeds for all larger subsets. We reduce the number of seeds for larger subsets because the inherently lower variance in the validation accuracy doesn't justify paying the computational cost associated to fine-tuning more seeds.

**Target datasets.** We transfer our models to a broad set of target datasets, including (1) CIFAR-100, (2) CIFAR-10, (3) SVHN, (4) Fashion MNIST (Xiao et al., 2017), (5) KMNIST and (6) MNIST. Since all of these datasets have images at a lower resolution than ImageNet, we up-scale our images with bi-linear interpolation. In addition, we use common data transform techniques such as random cropping and rotation that are well-known to produce high-quality results with certain datasets.

### A.4 FINE-TUNING DETAILS

Figure 9 illustrates all four fine-tuning configurations in our experiments. Notice how in Subfigure 9(d) we unfreeze more than half of the ResNet-50 architecture, thereby testing what occurs as we fine-tune a lot of blocks.

All source models are fine-tuned to all datasets using stochastic gradient descent with momentum using the hyperparameters described in Table 3.

Table 3: Hyper-parameter summary for all fine-tuned source models

| Learning rate | Batch size | Momentum | Weight decay | LR decay | LR decay schedule | Fine-tuned adversarially? |
|---|---|---|---|---|---|---|
| 0.1 | 128 | 0.9 | $5 \times 10^{-4}$ | 10x | 1/3, 2/3 epochs | No |

The learning rate decays to a tenth of it's current value every 33 or 50 epochs, which corresponds to 1/3 of the total fine-tuning epochs, as shown in Table 4. Also, the test accuracy frequency refers to how often is the test accuracy computed, in epochs. So, for example, if the test accuracy frequency is 20, then we check the test accuracy after epoch 1, 21, 41, ..., 81, and 100.

With regards to the random seeds, we have the following formula to define the set of seeds used, $S_k$, as a function of the total number of random seeds used, $k$:

---

[1]The models that we use are provided as part of by the following repository: `https://github.com/MadryLab/robustness`.

Table 4: Batch summary for every target dataset and source model

| Number of images | Fine-tuning epochs | Number of random seeds | Test accuracy frequency (epochs) | LR decay schedule |
|---|---|---|---|---|
| 100 | 100 | 20 | 20 | 33/66 |
| 200 | 100 | 20 | 20 | 33/66 |
| 400 | 100 | 20 | 20 | 33/66 |
| 800 | 100 | 20 | 20 | 33/66 |
| 1,600 | 100 | 20 | 20 | 33/66 |
| 3,200 | 150 | 10 | 10 | 50/100 |
| 6,400 | 150 | 10 | 10 | 50/100 |
| 12,800 | 150 | 5 | 10 | 50/100 |
| 25,600 | 150 | 5 | 10 | 50/100 |
| All | 150 | 1 | 10 | 50/100 |

$$S_k = \{20000000 + (100000i)|i \in \{0, 1, \cdots, k-1\}\}. \tag{6}$$

Thus, when we use 20 seeds, as it is the case for the subset of 100 images, we use seeds 20000000, 20100000, ..., 21900000. Large numbers were used to avoid numerical instability issues that arise with small numbers where their binary representation has too many zeroes.

See Table 5 for additional detail with regards to the source models. Notice that although adversarially-trained models do worse on the source dataset, they outperform naturally-trained models on the target datasets, as shown in Table 7.

Table 5: Summary of source models trained on ImageNet, which we consider for transfer learning.

| Pre-training Procedure | Constraint | ImageNet Test Accuracy |
|---|---|---|
| Natural | – | 76.13% |
| Adversarial | $\|\delta\|_2 \leq 3$ | 57.90% |
| Adversarial | $\|\delta\|_\infty \leq 4/255$ | 62.42% |
| Adversarial | $\|\delta\|_\infty \leq 8/255$ | 47.91% |

See Table 6 for a high-level overview of all datasets used. This should serve as a reminder that our source dataset is ImageNet, with an extensive 1.2 million training images and 1,000 labels, it serves as a great starting point in our experiments. All other target datasets have a considerably lower number of training and test images.

Table 6: Summary of the source and target datasets.

| Dataset | Number of training images | Number of test images | Number of labels | Color or grayscale | Source or target |
|---|---|---|---|---|---|
| ImageNet | 1.2 million | 150,000 | 1,000 | color | source |
| CIFAR-100 | 50,000 | 10,000 | 100 | color | target |
| CIFAR-10 | 50,000 | 10,000 | 10 | color | target |
| SVHN | 73,257 | 26,032 | 10 | color | target |
| FMNIST | 60,000 | 10,000 | 10 | grey-scale | target |
| KMNIST | 60,000 | 10,000 | 10 | grey-scale | target |
| MNIST | 60,000 | 10,000 | 10 | grey-scale | target |

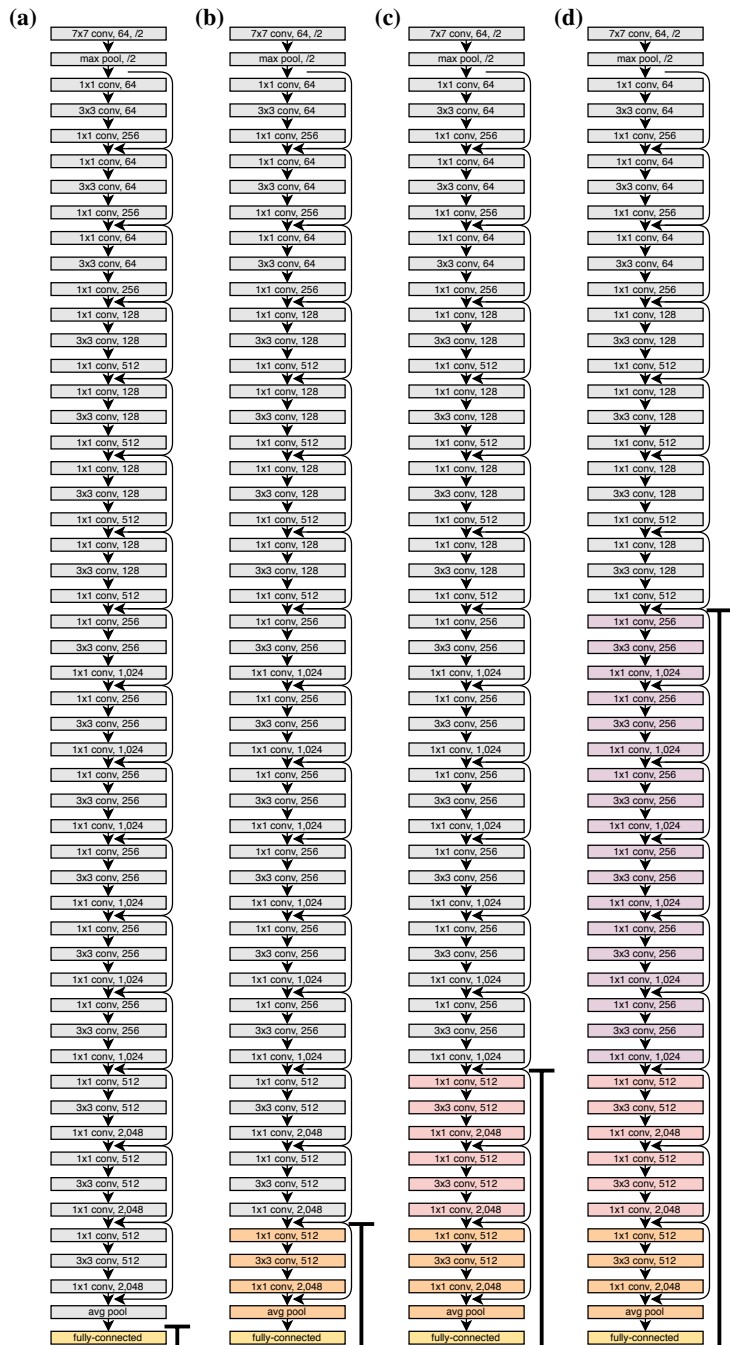

Figure 9: Fine-tuning setup: (a) Zero blocks. (b) One block. (c) Three blocks (d) Nine blocks. Each block has Three convolutional layers.

## A.5 ADDITIONAL RESULTS

Table 7 reports the test accuracy of all of our source models after fine-tuning three blocks using different numbers of training images on each of the six target datasets. The rightmost column shows the non-transferred model trained only on the target dataset and trained on the entire network. The average test accuracy is reported for all cases where the model is fine-tuned with less than the entire training set. The bolded numbers represent the highest test accuracy among source models. From this table, we can see that the robust models consistently outperform the natural models.

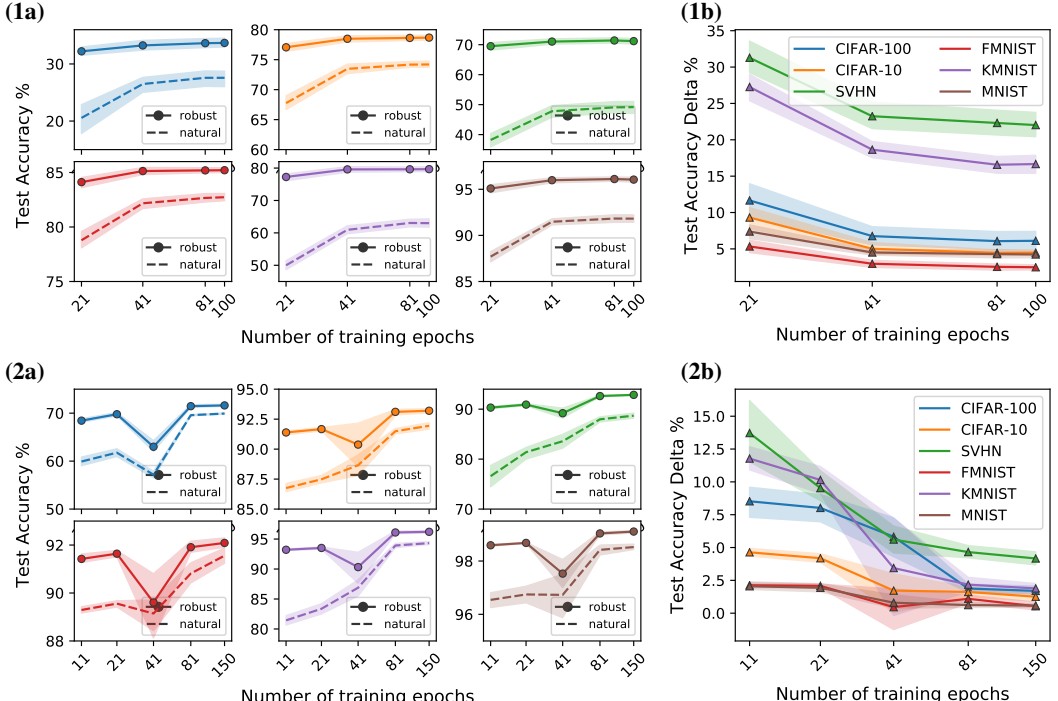

Figure 10: Robust models transfer faster. (a) Shows the test accuracy during the fine-tuning process on the six target datasets (color-coded as in (b)). (b) Shows the test accuracy delta, defined as the robust model test accuracy minus the natural model test accuracy. The solid line is the mean and its shade is the 95% confidence interval. Both the robust and natural models are ResNet50s' trained on ImageNet. The robust model uses a $\|\delta\|_2 \leq 3$ constraint. Both models fine-tune three convolutional blocks using a random subset of 800 images ($\sim 2\%$) in (1) and 12,800 images ($\sim 26\%$) in (2) of the target dataset.

## A.6 ADDITIONAL DETAIL ON LEARNING FASTER

The following subsection contains the additional charts that were omitted in Figure 3 in Section 4. Figure 11 shows the same behavior is observed in all three figures: It's sub-optimal to fine-tune either 0 or nine convolutional blocks, as opposed to one or three. Consistent with our methodology for Figure 4, we adversarially-train models on ImageNet and then fine-tune various numbers of convolutional blocks using a random sample of images in the target dataset.

## A.7 ADDITIONAL DETAIL ON THE EFFECT OF THE NUMBER OF FINE-TUNED BLOCKS

The following subsection contains the additional charts that were omitted in Figure 4 in Section 4. Figure 11 shows the same behavior is observed in all three figures: It's sub-optimal to fine-tune either 0 or nine convolutional blocks, as opposed to one or three. Consistent with our methodology for Figure 4, we adversarially-train models on ImageNet and then fine-tune various numbers of convolutional blocks using a random sample of images in the target dataset.

## A.8 COMPUTATIONAL COST

In general, a PGD($k$) adversarial training process as described by Madry, is $k$ orders of magnitude more expensive than natural training. This can be seen directly from the fact that there are $k$ more iterations in the inner maximization loop of the risk minimization procedure. However, since only the source model must be trained adversarially, and these source models can be downloaded from publicly available repositories, the marginal computational cost of fine-tuning to a target dataset is

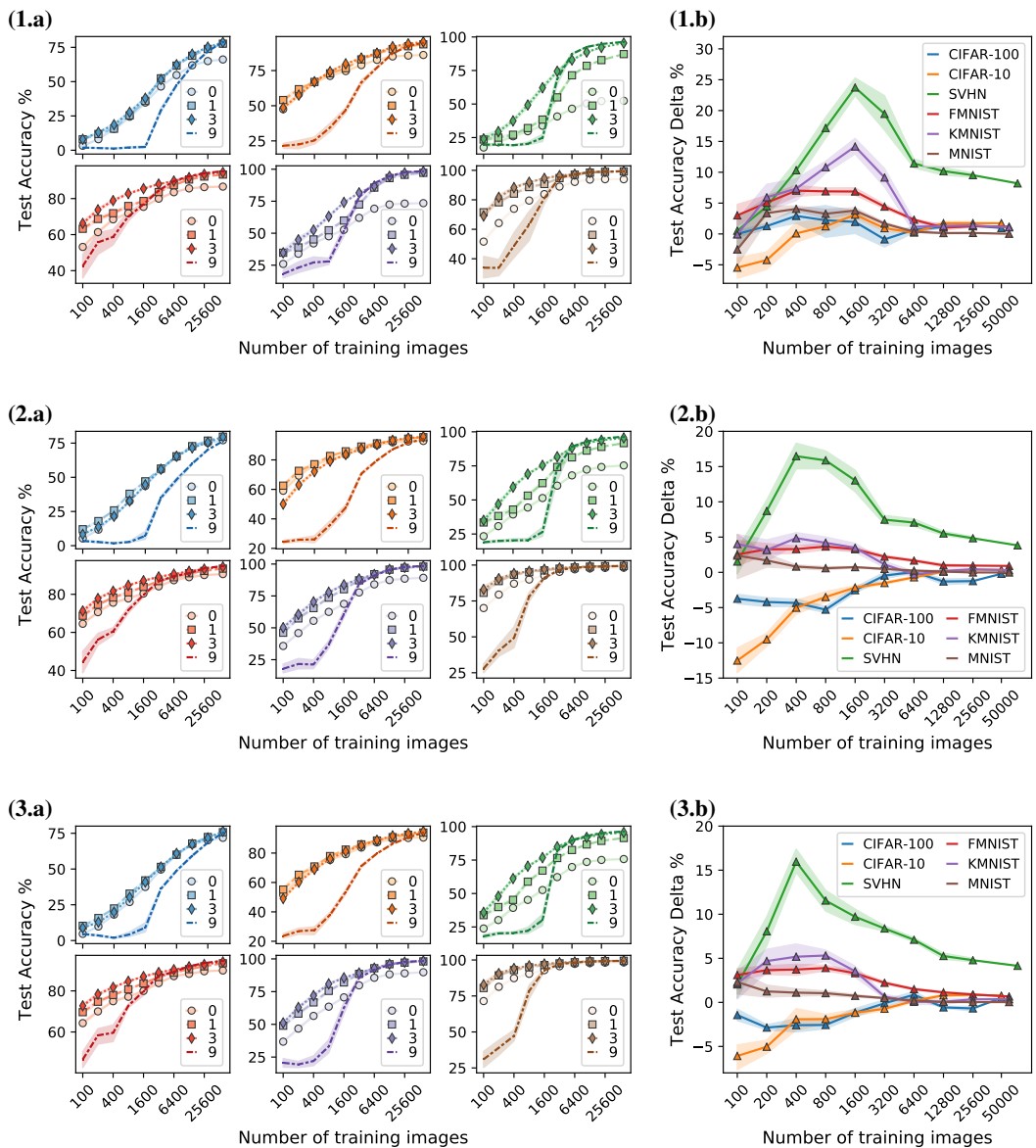

Figure 11: It's sub-optimal to fine-tune either zero (only FC layer) or nine convolutional blocks, as opposed to one or three. (a) Test accuracy on each of the six target datasets (color-coded as in (b)) for various subset sizes and fine-tuning various number of blocks: zero, one, three, or nine. (b) Test accuracy delta is defined as the test accuracy with three fine-tuned blocks minus the test accuracy with one fine-tuned block. Adversarial constraints used to train on the source dataset (ImageNet): (1) $\|\delta\|_2 \leq 3$, (2) $\|\delta\|_\infty \leq \frac{4}{255}$, or (3) $\|\delta\|_\infty \leq \frac{8}{255}$. The solid line is the mean and the shade is the 95% confidence interval.

perhaps more important. Fortunately, the computational cost of fine-tuning both robust or natural models is the same.

A.9 ADDITIONAL DETAILS ON INFLUENCE FUNCTIONS

Suppose that $\ell : \mathbb{R}^m \times \mathbb{R}^d \to \mathbb{R}$ is a smooth loss function and $x^1, \ldots, x^n \in \mathbb{R}^m$ are our given data. The empirical risk minimization (ERM) problem takes the form

$$\min_{\theta \in \mathbb{R}^d} f(\theta) = \frac{1}{n} \sum_{j=1}^n \ell(x^j, \theta). \tag{7}$$

Say $\theta^\star$ is the argmin solution of the above optimization problem. Let's now consider upweighing a data point $x_{\text{train}}$ with $\epsilon \in \mathbb{R}$. This modifies the learning problem as:

$$\min_{\theta \in \mathbb{R}^d} \frac{1}{n} \sum_{j=1}^n \ell(x^j, w) + \epsilon \ell(x_{\text{train}}, \theta). \tag{8}$$

Let $\theta_\epsilon^\star$ be the solution of the upweighted problem above. The *influence* of a training data point $x_{\text{train}}$ on a test data point $x_{\text{test}}$ approximates the change in the function value $\ell(x_{\text{test}}, \theta^\star) \to \ell(x_{\text{test}}, \theta_\epsilon^\star)$ with respect to an infinitesimally small $\epsilon$, i.e., when $\epsilon \to 0$ when $x_{\text{train}}$ is upweighed by $\epsilon$. This can be calculated in closed form Koh & Liang (2017) as:

$$-g_1 H^{-1} g_2, \tag{9}$$

where $g_1 = \nabla \ell(x_{\text{train}}, \theta^\star)^\top$, $g_2 = \nabla \ell(x_{\text{test}}, \theta^\star)$ and $H$ is the Hessian of the loss function $\nabla^2 f(\theta^\star)$. In particular, we first compute the Hessian of the source model fine-tuned with 3,200 CIFAR-10 images as the sum of the Hessians of the loss of batches of five images. Note that we only use the 3,200 images that were used in the fine-tuning process, since it accurately reflects the Hessian of the model. Then we get $H^{-1}$ using the (Moore-Penrose) pseudo-inverse, using its singular-value decomposition and including all singular values larger than 1e-20.

Koh et. al. Koh & Liang (2017) discuss optimization speedup techniques to determine the most influential $x_{\text{train}}$ for a given $x_{\text{test}}$ at scale. However, finding top-$k$ influential images is a combinatorial problem for $k > 1$. So, typically a greedy selection of the next top influential image is made iteratively $k$ times. Further, selecting multiple images also requires consideration of interaction and group effects. As such, the top-5 influential images are likely to be less representative of actual influence being asserted than one would expect.

**Fisher kernels and influence functions** Khanna et. al. Khanna et al. (2019) recently discovered an interesting relationship between Fisher Kernels and Influence functions: if the loss function $\ell(\cdot)$ can be written as a negative log-likelihood, then at the optimum $w^\star$, the Fisher dot product between two points is exactly the same as the influence of those points on each other (note that the influence is a symmetric function). In other words, finding the most influential data point to a given data point is equivalent to finding the nearest neighbor of the point in the space induced by the Fisher kernel. As observed in Section 6 for robust training, most influential points for a data point tend to be largely the ones belonging to the same label. This implies that the in the Fisher space, the points with the same label tend to be grouped together.

A.10 ADDITIONAL DETAILS FOR ADVERSARIAL ATTACKS COMPARISON SECTION

Both the PGD(1) and Gaussian models are ResNet-50's trained on ImageNet-1K using stochastic gradient descent with momentum and the following hyperparameters: 0.1 learning rate, 128 batch size, 0.9 momentum, 10x learning rate decay, and an equally-spaced (i.e. linear) learning rate decay schedule. The test accuracy of each of these models is 60.29% and 74.02% for the PGD(1) and Gaussian model, respectively. Both models use the $\|\delta\|_2 \leq 3$ adversarial constraint.

The PGD(1) model uses one attacker step, with a step size of 6 and the perturbation is initialized at zero. The Gaussian model adds a perturbation for each pixel drawn from a standard Normal distribution $\mathcal{N}(\mu = 0, \sigma^2 = 1)$.

A.11 CODEBASE OVERVIEW

The starting point requires downloading the source ImageNet models, and installing the appropriate libraries. Next, the user can decide how to fine-tune the source models: individually or in batches.

The `train.py` file will allow individual training, while the `tools/batch.py` file allows training in batches.

The `train.py` file contains 9 parameters that are explained by running the following command: `python train.py --help`. Also, the `helpers.py` and `delete_big_files.py` files under the `tools` folder contain the logic that supports the `train.py` file. This includes the random subset generator, the fine-tuning procedure, and the data transforms.

Separately, note that when running the `batch.py` file, the fine-tuned models won't be saved into the `results/logs` directory. This is due to the fact that models can occupy a significant amount of memory and we do not plan to use these fine-tuned models in the future. However, if the user wants to save the fine-tuned models, then he or she can do so by commenting our line 60 in the `batch.py` file: `deleteBigFilesFor1000experiment()`.

Lastly, all results are stored into the `results/logs` folder by default and can be compiled easily into a csv file using the `log_extractor.py` script.

## A.12 DETAILED FUTURE WORKS

Even though we support our main thesis with extensive empirical evidence and analyze this phenomenon through the lens of texture bias and influence functions, why, when, and how robust models transfer better deserves further investigation. In this section we'd like to provide some ideas that we hope will spark research interest.

**Different adversarial training constraint type.** Prior work only considers $\ell_2$ and $\ell_\infty$ adversarial constraint types. However, different adversarial constraints could allow models to retain more transferable features from the source dataset. Two possibilities are (i) constraining on the Fischer Kernel with influence functions, and (ii) constraining on the Fourier space instead of the pixel space. For (i), as shown by Koh & Liang (2017) for the $i^{th}$ image $x_i$ we can compute the perturbation $\delta_i$ at each step of PGD by starting with $\delta_i = 0$, and then $\delta_i = \mathcal{P}(\delta_i + \alpha \mathrm{sign} \mathcal{I}_{pert,loss}(x_i + \delta_i, x_i))$. For (ii), we could instead calculate the gradient w.r.t. each one of the frequencies and constrain the model to only use a subset of all of its frequencies to represent the input image.

**Different source datasets.** ImageNet might not be the best source dataset for two reasons. First, we as shown by Tsipras et al. (2020) there are many labels that overlap with each other, such as rifle and assault rifle. Second, there are many training images containing objects from more than one label, referred to in Tsipras et al. (2020) as multi-object images. Thus, we think it would be worthwhile to use Tencent's Large-Scale Multi-Label Image Database from Wu et al. (2019) as a source dataset instead of ImageNet.

**Decision-boundary bias.** In line with Section 5, it might be worth looking at the transferability of robust models as a function of how closely related are the labels in the target dataset. Our hypothesis is that if the labels in the target dataset are closely related, then the robust model might transfer slightly worse than if the labels were further apart from each other. Although measuring the closeness of labels within a dataset is challenging, this could be an interesting extension to Santurkar et al. (2020).

**New use-cases.** As shown in Section 5, robust models are biased towards low resolutions and low frequencies. Thus, it's possible that robust models have a lower facial recognition bias than naturally trained models.

Table 7: Summary of the test accuracy on target datasets after fine-tuning three blocks (nine convolutional layers) except for the rightmost column, which shows the non-transferred model trained on the entire network. Reported average test accuracy for all cases where the model is fine-tuned with less than the entire training set.

| Target dataset | Training images | Pre-training constraint on source dataset (ImageNet) | | | | Trained From Random Init |
|---|---|---|---|---|---|---|
| | | Natural | $\|\delta\|_2 \leq 3$ | $\|\delta\|_\infty \leq \frac{4}{255}$ | $\|\delta\|_\infty \leq \frac{8}{255}$ | |
| CIFAR-100 | 100 | 7.97 | 8.54 | 8.24 | **8.84** | 1.34 |
| CIFAR-10 | 100 | 48.49 | **51.89** | 49.86 | 49.03 | 12.29 |
| SVHN | 100 | 23.5 | **37.9** | 34.75 | 35.75 | 19.14 |
| FMNIST | 100 | 66.11 | 71.8 | 71.24 | **72.69** | 19.91 |
| KMNIST | 100 | 34.29 | **51.76** | 50.12 | 51.05 | 12.42 |
| MNIST | 100 | 68.69 | **83.83** | 83.03 | 82.81 | 12.79 |
| CIFAR-100 | 200 | 13.05 | 13.26 | **13.71** | 12.79 | 1.35 |
| CIFAR-10 | 200 | 57.44 | **64.05** | 62.83 | 60.11 | 11.01 |
| SVHN | 200 | 29.37 | **50.89** | 46.82 | 47.77 | 19.16 |
| FMNIST | 200 | 73.53 | **78.51** | 77.67 | 78.33 | 19.88 |
| KMNIST | 200 | 44.77 | **62.97** | 60.36 | 62.84 | 12.63 |
| MNIST | 200 | 81.58 | **90.72** | 90.44 | 90.49 | 15.66 |
| CIFAR-100 | 400 | 18.57 | **21.74** | 21.32 | 19.79 | 1.36 |
| CIFAR-10 | 400 | 66.84 | 71.65 | **71.93** | 68.81 | 12.01 |
| SVHN | 400 | 37.3 | **61.97** | 58.99 | 61.02 | 19.36 |
| FMNIST | 400 | 78.8 | **82.09** | 81.4 | 81.58 | 35.23 |
| KMNIST | 400 | 52.45 | 71.65 | 70.18 | **71.89** | 13.13 |
| MNIST | 400 | 88.09 | **94.28** | 93.9 | 93.98 | 22.57 |
| CIFAR-100 | 800 | 27.57 | **33.68** | 32.55 | 30.17 | 2.13 |
| CIFAR-10 | 800 | 74.2 | 78.68 | **78.99** | 75.95 | 20.86 |
| SVHN | 800 | 49.17 | **71.19** | 68.84 | 70.19 | 19.51 |
| FMNIST | 800 | 82.73 | **85.21** | 84.6 | 84.59 | 61.12 |
| KMNIST | 800 | 63 | 79.67 | 77.73 | **80.68** | 16.86 |
| MNIST | 800 | 91.81 | **96.05** | 95.58 | 95.92 | 38.6 |
| CIFAR-100 | 1,600 | 37.88 | **45.64** | 44.49 | 40.82 | 5.13 |
| CIFAR-10 | 1,600 | 79.74 | **83.76** | 83.47 | 81.1 | 30.98 |
| SVHN | 1,600 | 62.14 | **77.84** | 75.24 | 76.77 | 19.6 |
| FMNIST | 1,600 | 85.39 | **87.58** | 87.09 | 86.91 | 78.14 |
| KMNIST | 1,600 | 74.06 | 84.95 | 83.66 | **85.52** | 30.13 |
| MNIST | 1,600 | 94.47 | 97.13 | 97 | **97.16** | 85.55 |
| CIFAR-100 | 3,200 | 51.46 | **56.91** | 55.79 | 51.34 | 20.94 |
| CIFAR-10 | 3,200 | 83.77 | **87.56** | 87.34 | 85.07 | 55.99 |
| SVHN | 3,200 | 74.31 | **85.55** | 81.57 | 84.4 | 22.94 |
| FMNIST | 3,200 | 87.97 | **89.43** | 89.11 | 88.89 | 83.47 |
| KMNIST | 3,200 | 81.42 | **89.68** | 88.26 | 89.14 | 81.51 |
| MNIST | 3,200 | 96.4 | **98.13** | 97.93 | 98.1 | 96.89 |
| CIFAR-100 | 6,400 | 62.26 | **66.1** | 65.31 | 60.75 | 36.74 |
| CIFAR-10 | 6,400 | 87.28 | **90.91** | 90.23 | 88.6 | 71.37 |
| SVHN | 6,400 | 82.63 | **90.46** | 88.17 | 89.09 | 83.87 |
| FMNIST | 6,400 | 89.63 | **90.73** | 90.44 | 90.11 | 87.41 |
| KMNIST | 6,400 | 86.7 | **93.61** | 91.7 | 92.86 | 90.88 |
| MNIST | 6,400 | 97.34 | **98.73** | 98.5 | 98.57 | 98.46 |
| CIFAR-100 | 12,800 | 69.92 | **71.61** | 71.42 | 67.16 | 52.32 |
| CIFAR-10 | 12,800 | 91.95 | **93.2** | 93.11 | 91.6 | 81.19 |
| SVHN | 12,800 | 88.7 | **92.86** | 91.68 | 91.81 | 92.44 |
| FMNIST | 12,800 | 91.56 | **92.09** | 91.97 | 91.69 | 90.32 |
| KMNIST | 12,800 | 94.32 | **96.21** | 95.73 | 95.9 | 95.7 |
| MNIST | 12,800 | 98.52 | 99.11 | 98.98 | 99.03 | **99.27** |
| CIFAR-100 | 25,600 | 75.08 | **75.68** | 75.4 | 71.52 | 67.75 |
| CIFAR-10 | 25,600 | 93.93 | **94.92** | 94.63 | 93.28 | 89.92 |
| SVHN | 25,600 | 92.24 | 94.36 | 93.76 | 93.91 | **94.93** |
| FMNIST | 25,600 | **93.42** | 93.24 | 93.14 | 92.79 | 92.25 |
| KMNIST | 25,600 | 97.01 | 97.52 | 97.34 | 97.44 | **97.96** |
| MNIST | 25,600 | 99.06 | 99.27 | 99.21 | 99.24 | **99.53** |
| CIFAR-100 | 50,000 | 78.49 | 79.24 | **79.51** | 76.06 | 79.42 |
| CIFAR-10 | 50,000 | 95.36 | **95.86** | 95.6 | 94.58 | 94.08 |
| FMNIST | 60,000 | 94.51 | **94.72** | 94.41 | 93.98 | 93.97 |
| KMNIST | 60,000 | 98.07 | 98.35 | 98.19 | 98.38 | **99.01** |
| MNIST | 60,000 | 99.19 | 99.42 | 99.34 | 99.39 | **99.68** |
| SVHN | 73,257 | 95.33 | 96.02 | 95.42 | 95.63 | **96.53** |

