# OpenReview forum: "Adversarially-Trained Deep Nets Transfer Better: Illustration on Image Classification"
_ICLR.cc/2021/Conference — ICLR 2021 Poster_

### Official Review · AnonReviewer2 · 2020-10-21
**Review for "Adversarially-Trained Deep Nets Transfer Better"**

**Rating:** 7
**Confidence:** 4

**Review:**

### Contributions ###
* The paper proposes that models that were adversarially trained transfer better to other datasets in that they increase _clean_ performance on this target dataset if there are only few labeled datapoints for the target task or only few training epochs are conducted
* The authors test their hypothesis for ResNets pretrained on Imagenet in different threat models and transfer these models to 6 different target datasets. Generally, results provide sufficient evidence for the paper's main hypothesis (robust models transfer better)
* Additional experiments provide evidence that better transferability of robust models is partly due to relying more on shape rather than texture cues. Moreover, an additional analysis using influence functions leads to the hypothesis that  robust neural networks might have learned to classify using example-based concept learning like in human beings.

### Significance ###
Transfer learning is a topic of high relevancy for practitioners since it can reduce both data label effort and training time. Improving upon the baseline of transferring models pretrained on Imagenet with a non-adversarial loss is thus a potential significant result.
However, the paper's review of the transfer learning literature is superficial and misses some relevant related work such as "Rethinking ImageNet Pre-training" by He et al., ICCV 2019.  Additionally, Geirhos et al. (ICLR 2019) also showed that models pretrained on stylized ImageNet (and thus having a stronger shape bias) transfer better to object detection tasks. This should be mentioned in Section 5.  And more generally: if stronger transferability is mainly due to increased shape bias, wouldn't it make sense to pretrain explicitly for strong shape bias rather than achieving this indirectly via adversarial training as proposed in this paper?
A more thorough review of the transfer learning literature and relating the obtained results to this would generally strengthen the paper.

### Originality ###
The work is a purely empirical work studying the stated hypotheses, no novel methods are proposed. Originality can thus only come from the hypotheses.
The main hypothesis (robust models transfer better) was also proposed by Salman et al. (2020). However, this work should be seen as concurrent since it was released on arXiv only four months ago.
The main prior work is Shafahi et al. (ICLR 2020), which also studies transferring adversarially pretrained models to other tasks. However, their focus is on the robustness gains of transferred models rather than on the effect on clean performance.
In summary, I think the main hypothesis studied in the paper is original. However, it is also clearly only a relatively small incremental step beyond what Shafahi et al. 2020 did.

### Clarity ###
Experimental setup, training pipeline, and analysis are outlined clearly. Releasing code for finetuning and replicating the experiments would further strengthen reproducibility

### Quality ###
Generally, the experiments are well conducted, covering a broad range of threat models, target datasets, training image and epochs regimes, and finetuning strategies. Additionally, Section 5 and 6 shed additional light onto why robust models might transfer better, and by this further strengthening the main message of the paper.
One shortcoming is that all target tasks are image classification tasks. Whether robust models also transfer better to task such object detection or semantic segmentation remains unclear.

### Recommendation ###
In summary, I think the paper is a nice experimental study of a clearly stated hypothesis with potential practical impact. I thus lean towards acceptance, even though novelty is clearly borderline.

### Final Recommendation after Author Response###
The authors have addressed several of my main concerns. It would have been helpful to study transferability to tasks beyond image recognition, but overall, I think the paper has been considerably improved. I increase my score to 7.
Two remarks regarding new content:
 * I find it misleading to denote PGD as a targeted adversary and additive Gaussian noise as a random adversary Section 7. "Targeted" usually refers to an adversary that aims at achieving a specific misclassification (target class). Gaussian noise is not really an adversary, rather a distortion/image corruption. I would recommend clarifying the naming to avoid confusion of readers.
 * Is there any particular reason to use PGD(3) in Table 1b for evaluation? Would the effect hold also against stronger attacks (more iterations etc.)?

---

> ### Author Response · Authors · 2020-11-17
> **Making the case for more than an incremental step beyond Shafahi et al. 2020 and also clarifying our scope**
>
> Thank you for the positive feedback and constructive comments.
>
>
> **Although the main hypothesis is original, it is only a small incremental step beyond Shafahi et al. 2020.**
>
> We strongly disagree that the results in our paper can be seen as only a small incremental step. In fact, our results contradict Shafahi et al. (ICLR 2020) results. We show that robust models transfer better, while Shafahi et al. (ICLR 2020) show that robust models transfer worse: "an ImageNet robust model with a ∥δ∥₂ ≤ 5 constraint has lower accuracy on the target datasets, CIFAR-10 and CIFAR-100, compared to a natural ImageNet model."
>
> We hope the reviewer agrees that since our conclusion is diametrically opposite, it should be shared with the community. An additional contribution of our work is looking into the transfer learning behavior with less data, which was previously unexplored and quite valuable from a practical standpoint. Furthermore, we explain, at least partly, why robust models transfer better, with the use of influence functions and shape bias. And to the best of our knowledge, this approach to understanding the transfer performance of models is novel.
>
> **Not clear if robust models also transfer better on tasks different from image classification.**
>
> We will clarify in our abstract and introduction that our focus is only on image classification tasks. Moreover, we're also willing to change our title to "Adversarially-Trained Models Transfer Better On Image Classification" if our paper gets accepted. Separately, we believe that this approach will also work on a broader range of tasks, such as image segmentation, which future works could explore.

---

> > ### Comment · AnonReviewer2 · 2020-11-18
> > **Further Clarification**
> >
> > Thanks for the feedback. I did acknowledge in my review that the author's study and findings are different to Shafahi et al. (ICLR 2020), so there is no strong disagreement here. However, I would maintain my stance that both works are closely related and the author's work builds upon Shafahi et al. (ICLR 2020) in studying transferability of adversarially trained model. The novelty lies in studying transferability to  novel tasks in terms of clean performance and in terms of number of labeled datapoints in target task. This makes it in my assessment an incremental, however sufficiently novel contribution.
> >
> > I would like to ask the authors two questions that would be helpful for the final review:
> >  * The authors state that their results contradict  Shafahi et al. (ICLR 2020): "We show that robust models transfer better, while Shafahi et al. (ICLR 2020) show that robust models transfer worse". What is the authors explanation for this disagreement? (if it is discussed in the paper, could the authors point me to the respective section?)
> >  * Repeating my question from the review: "if stronger transferability is mainly due to increased shape bias, wouldn't it make sense to pretrain explicitly for strong shape bias rather than achieving this indirectly via adversarial training as proposed in this paper?" To make this more specific: did the authors also consider evaluating the transfer properties of the model from Geirhos et al. "ImageNet-trained CNNs are biased towards texture; increasing shape bias improves accuracy and robustness"that was trained on stylized ImageNet?

---

> > > ### Author Response · Authors · 2020-11-19
> > > **Explanation for the contradiction with Shafahi et al. (2020) and transferring the stylized ImageNet model**
> > >
> > > We appreciate the thoughtful response and that you recognize the novelty of some of our contributions. Additionally, we’d like to highlight two additional novel contributions. First, inspired by R4, we added a new section studying if other adversarial attacks improve transferability (PGD(1) and random Gaussian noise). Second, we explain, at least partly, why robust models transfer better, with the use of influence functions and shape vs. texture bias.
> > >
> > > **What is the author's explanation for this disagreement with Shafahi et al. (2020)?**
> > >
> > > The goal of Shafahi et al. (2020) is different from ours. They want to show that robust models are still resilient to adversarial attacks after fine-tuning (i.e., robustness transfers), that is why we believe that they chose larger perturbation bounds. In contrast, our goal is to show that robust models transfer better, which is why we chose smaller perturbation bounds. The different choice of perturbation bound size led to the disagreement, i.e., Shafahi et al. (2020) concluded that robust models transfer worse, while we conclude that robust models transfer better.
> > >
> > > We have already clarified this discrepancy in the related works section: “It might seem to contradict our thesis that they also notice that an ImageNet robust model with a $\|\delta\|_\infty \leq 5$ constraint has lower accuracy on the target datasets, CIFAR-10 and CIFAR-100, compared to a natural ImageNet model. However, we show that the robust model transfers better than the natural one when we use a $\|\delta\|_2 \leq 3$ constraint to adversarially train the source model.”
> > >
> > > And later, in Figure 5(c), we provide evidence that source models trained with a smaller adversarial constraint outperform source models trained with a larger one: The source model trained with $\|\delta\|_\infty \leq 4/255$ outperforms the one trained with $\|\delta\|_\infty \leq 8/255$, on CIFAR-10 and CIFAR-100.
> > >
> > > **Did the authors consider transferring the model from Geirhos et al. trained on stylized ImageNet?**
> > >
> > > Our original motivation was to present and study an unintended but positive side-effect of adversarial training, not to perform an extensive search to find the best pre-training process for transfer learning. Motivated by your question, we ran this experiment, but preliminary results showed that the model trained on stylized ImageNet does not transfer as well as the robust model. These experimental results agree with Salman et al. (2020). However, if you think that this is an important addition, we’re happy to add a sentence in the shape bias section, with detailed results in the appendix. What are your thoughts on this?

---

> > > > ### Comment · AnonReviewer2 · 2020-11-19
> > > > **On shape bias and the choice of epsilon**
> > > >
> > > > Thanks for the quick response. I understand that the paper is mainly about "an unintended but positive side-effect of adversarial training". But part of this also (a) what causes this side-effect (shape bias)? and (b) what follows from this observation?
> > > > And I think there is still room for clarification:
> > > >  * is it actually increased shape bias that causes increased transferability? If a model trained on stylized ImageNet does not transfer as well, this seems questionable.
> > > >  * if transferability increases with smaller epsilon, but epsilon=0 (clean training) is also suboptimal, how should one pretrain models adversarially to ensure one benefits from increased transferability? At the moment, this benefit seems to be quite brittle and dependent on the model. Can the authors make a clear point that transferability is dependent on epsilon and not on randomness in training, that is: do models trained with the same epsilon transfer equally well?

---

> > > > > ### Author Response · Authors · 2020-11-23
> > > > > **Shape bias does improve transferability, how to pretrain adversarially, and why our results are not the result of randomness**
> > > > >
> > > > > Thanks for the follow-up. Please see a response to your questions below:
> > > > >
> > > > > **Does shape bias improve transferability?**
> > > > >
> > > > > Yes. We have evidence to support the fact that the model trained on both stylized imagenet and imagenet imagenet is slightly *more transferable than the natural model but less transferable than the adversarially trained one*, in the low-data regime. This is consistent with our previous comment, and thus we see evidence that shape bias helps improve transferability.
> > > > >
> > > > > Below is a table with the mean accuracy for the natural, texture-robust, and adv-robust models, showing only the ones with statistically significant (95% confidence interval) differences between the natural and texture-robust models for 800 training images in the target dataset:
> > > > >
> > > > > |                 | Natural | Texture-Robust | Adv-Robust |
> > > > > |------------|:-------------:|:-------------:|:-------------:|
> > > > > | SVHN      | 49.2 | 55.6 | *71.2* |
> > > > > | KMNIST  | 63.0 | 70.2 | *79.7* |
> > > > > | MNIST    | 91.8 | 94.7 | *96.1* |
> > > > >
> > > > >
> > > > > **Effect of epsilon on transferability vs other randomness in training**
> > > > > Table 7 in the appendix shows a larger set of models and constraint setups, and it consistently shows the effect of epsilon on transferability. Further, if the reviewer is interested in deeper study of the effect epsilon has transferability, please refer to Figure 5(b) [A]. All of these experiments show that better transferability is dependent on choosing a good epsilon (which is a hyperparameter, and hence can be different for different datasets), and not on just other random variations in the training process.
> > > > >
> > > > > [A] https://arxiv.org/pdf/2007.08489.pdf

---

### Official Review · AnonReviewer4 · 2020-10-28

**Rating:** 6
**Confidence:** 3

**Review:**

1, Summary of contribution:
This paper claims that the pre-trained model trained adversarially can achieve better performance on transfer learning, and conducted extensive experiments on the efficacy of the adversely trained pre-trained models.
Also, the paper conducts an empirical analysis of the trained models and shows that the adversarially pre-trained models uses the shape of the images rather than the texture to classify the images.
Using the influence function (Koh 2017), the paper reveals that each influential image on the adversarially trained model is much more perceptively similar to its test example.


2, Strengths and Weaknesses:
The paper is well-written and organized, and the experiments look fair and well support the claim.   The analysis is interesting and insightful.
Meanwhile, the transfer is done to the domain of lower complexity, and some important comparative ideas are not extensively investigated.


3, Recommendation:
While the paper’s empirical results are solid, there seems to be a substantial room left for comparative studies.  More ablation studies shall be done for other regularization methods.
I believe that the paper is marginally above the acceptance threshold.

4, Reasons for Recommendation:
The reader will benefit more from the paper if the authors can justify their use of adversarial training as the regularization in the pretraining process.  I believe that this research warrants some comparative study for dropout, weight decay, as well as random perturbations.  I think the paper can be more insightful if it shows whether the other classical regularization performs better or worse on transfer learning than the proposed approach.

5,  Additional feedback:
In addition to the suggestions made in 4, I also believe that comparison shall be made against the model trained without pretraining.

\
---Post rebuttal---

Thank you for the response, and thank you for checking the performance comparison against the white-noise perturbation. It would be interesting to see a future work involving means other than Adversarial training (e.g. including other simple mechanisms like weight decay and dropout) to help reduce the overfitting effects in the pretraining phase. I would like to keep my score as is.

---

> ### Author Response · Authors · 2020-11-17
> **Comparing our approach (PGD(20)) to other robust training approaches and adding results for models trained from scratch with random initialization**
>
> Thank you for the positive feedback and constructive comments. We will add a new section to the main paper and add a column to Table 7 in the Appendix. Please see our comments below for more details.
>
> **Do source models trained with PGD(20) outperform other robust training approaches?**
>
> This is an excellent question, and to answer this, we are adding a new Section to the updated manuscript. To answer this question, we trained two ImageNet models: One with random Gaussian noise and another one with PGD(1). Our results show that PGD(20) and PGD(1) are similar to each other, and significantly better than Gaussian perturbations.
>
> **Can you compare your results to the model trained from scratch?**
>
> Yes, we will add a column to Table 7 in the appendix for reference but prefer to keep these results outside of the main paper. Here's our rationale: We know that both natural and robust transferred models will massively outperform models trained from scratch with random initialization, especially as the number of training images in the target dataset decreases. Showing this comparison in our figures in Section 4 would obscure our research's focus, which is not to show that transfer learning works, but that robust models transfer better than natural ones.

---

### Official Review · AnonReviewer3 · 2020-10-29
**Nice and clear with interesting experiments, but lacking context, a bit over-claimed, and a few other potential issues I'm optimistic can be addressed**

**Rating:** 6
**Confidence:** 4

**Review:**

**Summary of paper:** This paper investigates how "robust" (adversarially trained) models can improve the transfer of representations, finding that they transfer better. Additionally, they investigate some reasons this could be the case, examining the biases robust models appear to induce.

**Pros/strong points:**
 - interesting, well-explained experiments with mostly clear nice figures
 - nice extra investigations giving insight into the bias(s) conferred by adversarial training

**Cons/weak points:**
 - overstatement as though the results apply to any type of data/model, but only image data and convolutional nets are tried
 - potential issue with influence function experiments
 - no analysis of computational cost of adversarial training or other information on tradeoffs
 - background lacking/potential issues with related work

**Summary of review + recommendation:** Good paper with nice, thoughtful experiments, mostly well written, although I think the biggest issue there is over-claiming results applying to all data when only image data/convnets are studied. I'm also a bit worried about the thoroughness of the background research, and would like to see an analysis of the computational cost of adversarial training. If these and my question about influence function experiments are addressed, I would be happy to raise my score.

DETAILED REVIEW:

**Quality:**
 - Generally well written and organized, although the link between successive sections could be made a bit more strongly / flow could be better
 - Overstatement of scope (transfer learning in general, when only image data is studied) combined with misstatements in the first parts (about the origins of transfer learning) makes me skeptical of the depth of background research done and makes me suspect there may be some very relevant things which have not been surveyed/cited.  (see Specific questions/recommendations for details on both points, below)
 - Comparison of computational cost / other tradeoffs in adversarial vs. 'natural' training not discussed
 - Potential weakness/misleading conclusions in influence function experiments if I've understood correctly (see specifics below)

**Clarity:**
 - The abstract and introduction would benefir from more technical, specific language to avoid ambiguity and establish flow of the sections
 - Clarity within sections is pretty good., some specific suggestions below.
 - Figures are mostly nice and clear, not so much Table 1 and 7a though.

**Originality:**
 - This is the largest potential problem that I am most unsure about. I'm very familiar with work on statistical learning theory and generalization overall, but I'm not an expert in transfer learning or adversarial method and I'm not sure how well these works are reviewed, so I'm not sure how novel this work is. The experiments are well done and well explained though, and I think this is a good contribution even if it is less original than it is made to seem due to the lack of context.

**Significance:**
 - Nice insights and interesting experiments for those wanting to understand the impact of robust training on transfer
 - Limited practical insights without an analysis/discussion of tradeoffs e.g. overall computational time and stability.
 - Directions proposed for future work are concrete and interesting


**Specific questions/recommendations:**
 - the first sentence of abstract and the title talk about DL generically, but the second-sentence is about images specifically. If you add non-image data, I'd suggest rephrasing to make the 2nd sentence generic, and maybe mention this technique has been particularly successful for images. However since all experiments are with images, I'd suggest making the title and abstract specific to that domain. e.g. "Evidence from image data that adversarially-trained deep nets transfer better" or "Adversarially-trained deep nets transfer to new images better"
 - Overstatement of results: If you want the strong/general claims about transfer learning, I would strongly recommend doing experiments with at least MLPs in addition to convnets, and at least one other type of data in addition to  images. Otherwise the statements in title/abstract/intro should be circumscribed to more accurately reflect the nature of the experiments.
 - mention what is adversarially vs. naturally trained. although I strongly suggest using a different term than 'naturally', including in diagrams/elsewhere as it is confusing. It makes it seem like you are comparing adversarial training to natural gradient methods.
 - first sentence putting data hogs in quotes misleadingly suggests that it's a commonly used phrase. Suggest removing this i.e. "they are known to require large..." (and suggest adding a reference which quantifies this rather than referring to hearsay).
 - "similarly, "stunning" is an opinion, suggest "remarkable" or something like "excellent" which can be derived from empirical results - Comparison of computational cost / other tradeoffs :even just a reference where this is done with a sentence summarizing those results / other tradeoffs, e.g. "it's usually at least 2x as slow and more likely to be unstable" or something like that might be enough, but I would prefer to see full training curves and computational cost numbers in appendix, with a line or two summarizing these in the main text.
 - Stating that Caruana (1995) proposed transfer learning seems incorrect to me. There was a NeurIPS workshop that same year on the subject, suggesting it was already an established term at that time. I don't know the full history, but from a quick google it seems to have been very common in psychology / education literature in the 70s and 80s, here's a book that talked about it in the context of ML in the 80s :https://pdfs.semanticscholar.org/b547/c5837bff9347dc76330a72fd7cbc517ee08c.pdf and here's Rich Sutton talking about it in 92: https://link.springer.com/content/pdf/10.1007/978-1-4615-3618-5.pdf
 - Related work:    - covariate shift could also mention risk extrapolation https://arxiv.org/pdf/2003.00688.pdf (how does the extrapolation done there differ from the adversarial training?)    - it seems like a lot of works on adversarial and contrastive training and the relationship to generalization are missing; I'm not an expert in this but starting way back hard negative mining and other contrastive methods (e.g. word2vec) have been used and their properties discussed
 - The first subheading in section 4 is the conclusion drawn from that paragraph: "Adversarially-trained models transfer better and faster.", but subsequent headings do not have the same 'syntax' (they are more like titles than conclusions to be drawn). I like the conclusion-as-title format; I find it very engaging and helpful for skimming especially since there are many experiments. But most of all I would strongly suggest that all titles have the same 'syntax' i.e. if you can't think of conclusions-as-titles for the other bold p headers (although I think you can and should!), I would recommend rephrasing this one to be like the others (e.g. Comparison of adversarial and non-adversarial transfer)
 - formatting of table 1, with captions both below and above the tables, is hard to read. Put it all above or all below.
 - Could have been a nice opportunity to investigate whether robust models are more or less susceptible to the types of bias people worry about in real-world image datasets (e.g. face recognition); maybe worth mentioning this in future work
 - Unless I'm mistaken, the experiments with influence functions do not distinguish the effect of performance from that of training (bias toward the "human prior"). To do so, the robust and natural methods would have to have the same accuracy (i.e. this might involve very early stopping of the robust method to match the natural model performance). Without this, the qualitative and quantitative results could just be due to the higher accuracy of the robust method, not the particular form of prior it induces.

---

> ### Author Response · Authors · 2020-11-17
> **Clarifying scope to avoid overstating results, adding computational cost analysis, addressing accuracy differences in influence function experiments, and improving related works**
>
> Thank you for such a detailed review and suggestions to improve our paper. We are incorporating several reviewer suggestions, as stated below.
>
> **Overstatement of results**
> * **Can you make the title and abstract more specific to image classification?** We will clarify in our abstract and introduction that our focus is only on image classification tasks and other transfer tasks are out of scope in this work. Moreover, we're also willing to change our title to "Adversarially-Trained Models Transfer Better On Image Classification" if our paper gets accepted.
> * **Can you perform experiments with MLPs if you want to make general claims on transfer learning?** By changing the abstract, as described above, we make it more precise that our focus is on state-of-the-art deep neural networks for image classification tasks.
>
> **Can you add an analysis of the computational cost of adversarial training?**
> Yes. We will add a sentence on the computational cost trade-off in the main text and a new appendix section.
>
> **Influence function experiments**
> * **Are the results in the influence function experiments driven by the accuracy differences?** We address this question in our paper: "This vast gap is not explainable solely from only ~5% difference in target test accuracy, shown in Table 7 in Appendix A.5."
> * **Can you control for accuracy between natural and robust models in the influence experiments with early stopping?** Early stopping would add another bias, making it more challenging to make a fair comparison. We feel that the stark quantitative difference in Figure 7, and top-1/top-3 statistics given only 5% difference in accuracy suffices to support the qualitative claim of human prior bias suggested by Engstrom et al. (2019).
>
> **Background lacking/potential issues with related work**
> * **On origins of transfer learning.** We will reword the sentence to avoid suggesting that Caruana proposed transfer learning, and add other relevant citations.
> * **Can covariate shift also mention risk extrapolation?** There are many ways to handle covariate shift, but we are mainly concerned with using transfer learning. Thus, even though risk extrapolation is one method to handle covariate shift, it's not related to transfer learning. Therefore, the relevance to our work  is unclear to us.
> * **Missing works on adversarial and contrastive training and the relationship to generalization** We could add many weakly related works on adversarial training, contrastive learning, and their relationship to generalization. They're quite interesting, and we're well aware of them, but we don't see a strong connection to our work. It does seem that you feel quite strongly about expanding the related works section. We have purposely included only closely related works so far. If you could provide us with specific references that you feel improve the exposition during the discussion period, we are happy to add and discuss these.
>
> **Can you add reducing bias in real-world image datasets to future work?**
>
> We looked into facial recognition bias, but we couldn't see a direct relation to this work. However, we sincerely appreciate the idea and plan to investigate it. Could you please propose a few works that you think are worthwhile considering? As we gain more clarity on this idea, we might add this to the future works section.
>
> **Can you use a word other than "natural"? It can be confusing with natural gradient methods.**
>
> We will change "naturally-trained" to "non-adversarially-trained" both in the abstract and in the beginning of the introduction to avoid confusion. However, we feel strongly about using the term "natural" and "naturally-trained" throughout the paper to be consistent with the two most directly related works by Shafahi et al. (ICLR 2020) and Salman et al. (2020), as well with other Madry-PGD adversarial works such as Engstrom et al. (2019), Ilyas et al. (2019), and Tsipras (2018).
>
> **Other issues**
> * We plan on removing the "data hog" reference.
> * We plan on placing subtable captions on top for consistency.
> * We plan on changing the titles of multiple subsections to reflect takeaways/conclusions.

---

> > ### Comment · AnonReviewer3 · 2020-11-22
> > **Response to response**
> >
> > * Thanks for the more specific name change; this addresses my concern about over-claiming.
> > * I understand the motivation for using the term "natural" if that is what is used in previous work. In the sentence of the intro where it says  "non-adversarially-trained (i.e., natural)" I suggest instead saying "non-adversarially-trained (henceforth denoted **natural** as in previous works e.g. \cite{engstrom, ilyas, tspiras}" to communicate that, or if space is a concern, simply bold the word "natural" (also suggest bolding the word "robust" where it is defined; this makes it easy to scan back to see what those terms mean if a reader forgets later).
> > * The suggestion about facial recognition (or other) bias I was making was not in a prior work; similar to your experiments on imagenet, I was suggesting you could train a model to recognize faces or something both naturally and robustly and investigate whether / how the robust model is more or less biased toward certain categories (e.g. many works have shown that classifiers are poorer at identifying people of colour and women (http://gendershades.org/ has some associated papers); it would be interesting to know if adversarial training helps or even hurts - I can imagine that a shape bias would be helpful for recognizing faces regardless of colour, but perhaps if the shape bias is too 'specific' to male faces it could actually worsen performance on female or unusually-shaped faces.

---

> > > ### Author Response · Authors · 2020-11-23
> > > **Adding fairness as a future direction and emphasizing that we use the term “natural” to be consistent with prior works in the introduction.**
> > >
> > > We’re glad that you feel that we’ve addressed your concerns. Also, we’re grateful for the additional context on face recognition. We’ve made the changes described below.
> > >
> > > * We changed the wording and the format in the intro, as you suggested.
> > > * We added a fourth bullet point to the future work section discussing fairness, which is expanded in the appendix to mention face recognition.

---

> ### Comment · AnonReviewer3 · 2020-11-22
> **Last remaining issue: distinguishing effect of training from that of bias/prior in influence function experiments**
>
> Thank you for quite thoroughly addressing concerns; I think the paper is looking good and I would recommend acceptance if this point is addressed.
>
> I appreciate the results of the analysis in Figure 7 and find them interesting, but I'm afraid I still don't understand how these results demonstrate what you claim - I don't understand why a relatively small difference in accuracy couldn't explain even a relatively large difference in top-k influential images. I don't see how this can be shown except by controlling for accuracy (and even doing so would not tell the full story, but it's a step at least). I also don't understand why early stopping would give an undesirable bias. I'm not suggesting to remove the existing experiments and replace them with early-stopped ones; the existing ones are certainly informative, but if you don't make some attempt to control for accuracy or compare models with similar accuracy, I think the claim about this should be dialed back.

---

> > ### Author Response · Authors · 2020-11-25
> > **Influence functions**
> >
> > The effect of early stopping resulting in an implicit regularization/bias is well-known. Inclusion of such an additional change in the model would make it hard to isolate the effect of robust training on the induced prior.
> >
> > It is unclear how to effectively control all the other hyperparameters to control only the accuracy and at the same time not bias the learnt representations and hence the induced prior. Further, even if this was possible somehow, if the difference in accuracy is small (close to 0), and the difference in top-k for a given k is say some x% (x will likely be smaller than the current top-k difference), we feel that qualitatively this singular experiment would not be a bigger evidence of the conclusion we arrive at by saying a relatively small difference of 5% in accuracy vs a such a stark difference in top-k influences especially for small k suggests that semantically more meaningful representations being built internally and is supportive of Engstrom et al's view. Recall that the representations we refer to above  are the frozen layers of the model which have been learnt on the source dataset and transferred to the target datasets.
> >
> > If the reviewer still feels we should scale back our claims, we are open to considering suggestions about this. Our current conclusion regarding learning the concepts of image classes in neural networks is (quote from the paper):
> >
> > ".. the robust neural network has learned the image labels by creating strong associations to semantically-similar examples (akin to example based concept learning in human beings) in its internal representations. Thus, reinforcing the human
> > prior bias hypothesis in robust representations observed by Engstrom et al. (2019)."
> >
> > Would it please the reviewer if we change it to "... creating stronger associations than non-robust neural network to semantically-similar examples ... Thus, supporting the human prior hypothesis ..." ? We are also open to other suggestions the reviewer might have.

---

### Official Review · AnonReviewer1 · 2020-10-31
**Good paper to understand that robust models transfer better, but can be better.**

**Rating:** 6
**Confidence:** 4

**Review:**

This paper tries to investigate and understand if and how adversarial training helps the models trained on the source domain transfer easier and faster to target domains. With extensive different configurations (such as fine-tuning strategies) in experiments, the authors show that robust models transfer better than natural models with less training data from the target domain. Also they demonstrate the intuition behind through experiments, such as capturing shapes than textures or using influence functions.

Strengths
- The idea is interesting and have a potential for impacts in the community.
- Extensive experiments and investigations how and when the robust models works better than natural models is good to demonstrate the main ideas of this paper.
- Paper is easy to understand.

Weaknesses
- Even though it was shown by the experiments, it might need to have more theoretical understanding why the robust models transfer better or have better representations than natural models.
- Even though it seems to provide some explanations, it lacks more thorough investigation why the specific configuration choices yield better performances than others.
- The presented dataset choices seem limited, which could limit its potential impacts and applications in real-world problems (see the comments below).

Detailed comments:
- If the shape is indeed more important than texture for human-like performance, is it possible make the model even works on par with the natural models?
- Why specific configuration works better than others, such as fine-tuning three conv. blocks and ∥δ∥2 ≤ 3?
- In CIFAR-100 and (especially) CIFAR-10, fine-tuning one conv. block is better than zero conv. block and why?
- The target domains except CIFAR-100 and CIFAR-10 are all digit datasets, so its application to real-world problems may be limited. How about using different and non-overlapping classes than those in the source domain in other image datasets as target domains, such as CALTECH-256? It could make the paper stronger.
- In Table 5, the accuracy differences seem larger than ~5% as written in the text.

Typo:
Page 3: nx‘on-negative -> non-negative

---

> ### Author Response · Authors · 2020-11-17
> **Explaining specific configurations, discussing shape bias experiment, and clarifying our scope**
>
> Thank you for the positive feedback and constructive comments. We've addressed all of your concerns below.
>
> **Why do specific configurations work better than others?**
>
> We address why the number of fine-tuned convolutional blocks and adversarial constraints affect the transferability of robust models.
>
> * Number of fine-tuned convolutional blocks: In Section 4, we say that: "In particular, even though all other datasets transfer better when fine-tuning one or three blocks, it seems that models transfer better to CIFAR-10 and CIFAR-100 when fewer blocks are fine-tuned, as shown in Figure 4(b). This suggests that because these datasets are close to ImageNet, fine-tuning of early blocks is unnecessary." Yosinski et al., (2014) support the last statement: "Transferability is negatively affected by … the specialization of higher layer neurons to their original task at the expense of performance on the target task". We will rephrase this statement in the updated manuscript to make it more precise.
>
> * Adversarial constraints: In Section 4, we say that: "... a larger perturbation would destroy low-level features, learned from ImageNet, which are useful to discriminate between labels in CIFAR-10 and CIFAR-100. Finally, for datasets that are most distinct from ImageNet (SVHN and KMNIST), we find that robustness yields the largest benefit to classification accuracy and learning speed, as seen in Figure 2(b) and Figure 3(b), respectively. These discrepancies are even more noticeable when smaller fractions of the target dataset are used."
>
> From our empirical results, we can draw similarities between transfer learning configurations and hyperparameter tuning. To a large extent, the correct configurations will depend on the situation, and it's often difficult to know which configuration will work best ahead of time. Thus, we explored the landscape of fine-tuning configurations to avoid incorrectly concluding that our method works just because a specific configuration has been chosen.
>
> **If the shape is more important than texture, can robust models outperform natural models?**
>
> We address this question in Section 5, where we find that adversarially-trained models are less sensitive to texture variations. The setup that we use consists of training both a natural and a robust model on ImageNet-1K and testing on Stylized ImageNet before and after fine-tuning. This allows us to observe what happens when shape is more important than texture, which addresses the situation described in your question. Our results for this experiment show that the robust model significantly outperforms the natural one. Hence, robust models can outperform natural ones when shape is more important than texture.
>
> **Adding a deeper theoretical understanding of why robust models transfer better.**
>
> In this work, we focus on conducting an empirical investigation of the largely unexplored phenomenon of robust models transferring better. To explain, at least partially, why this happens, we also study the effect of shape bias and influence functions (Sections 5 and 6). However, we agree that a more theoretical understanding of why robust models transfer better is useful, and we hope that our work motivates future theoretical work.
>
> **Can you use target datasets with different and non-overlapping classes relative to ImageNet?**
>
> Besides CIFAR-100 and CIFAR-10, we use two non-digit target datasets with non-overlapping classes: Fashion-MNIST (i.e., FMNIST) and Kuzushiji-MNIST (i.e., KMNIST). FMNIST and KMNIST have clothing and cursive Japanese character classes, which are not contained in ImageNet.
>
> Similar to R2 and R4, we strongly believe that our experimental setup is thorough. This is because we use six target datasets (where four of them are non-digit and non-overlapping with the source dataset), which represent a wide variety of domains. R2 agrees, saying that "Generally, the experiments are well conducted, covering a broad range of threat models, target datasets, training image and epochs regimes, and fine-tuning strategies". Also, R4 also agrees, saying that "the experiments look fair and well support the claim."
>
> **Clarifications**
> * Table 5 shows accuracy differences larger than ~5%: We will fix the link to the correct table, which should be Table 7. The accuracy on the target dataset is our frame of reference, not the accuracy on the source dataset.
> * We will fix the typo on page 3.
>
> Full citation: Jason Yosinski, Jeff Clune, Yoshua Bengio, and Hod Lipson. How transferable are features in deep neural networks? In Neural Information Processing Systems (NeurIPS), 2014.

---

### Author Response · Authors · 2020-11-18
**General response**

We appreciate all the feedback from all the reviewers because it allows us to refine and improve our work. We’ve added a new section to our paper, reworded our abstract and introduction, and are considering changing the title of our paper, as described below.

**New Section 7: Do other adversarial attacks improve transferability?**
* Added a new section to analyze the transferability of two new ImageNet models trained with PGD(1) and random Gaussian noise, as proposed by R4.
* This Section becomes an additional contribution, addressing the concern for R2 on our work being more than an incremental step beyond Shafahi et al. (2020).
* Added five related works in this section, addressing R3’s concern about related works.

**Re-wording of title, abstract and introduction to clarify scope.**

We’re considering changing our title to “Adversarially-Trained Models Transfer Better On Image Classification”, since R1, R2, and R3 raised concerns related to our scope. We would sincerely appreciate feedback on this subject.

**Work in progress.**

We are computing results for the non-transferred ResNet-50 models with randomly initialized weights on the target datasets, which will be included in Table 7 of the Appendix by Friday.

---

> ### Comment · AnonReviewer3 · 2020-11-22
> **Appreciate name change, suggest "in" instead of "on"**
>
> Whether for "Adversarially-Trained Models Transfer Better in Image Classification" or Adversarially-Trained Models Transfer Better in Image Recognition", I think the "in" reads a bit better, but no big deal either way.

---

### Author Response · Authors · 2020-11-23
**Update**

We’ve uploaded a new version where we added a new column for models trained from scratch on the target datasets to Table 7 of the appendix, as suggested by R4. Please note that the models trained from scratch do not freeze any convolutional blocks. Thus, it is a somewhat unfair comparison to the other four columns, which freeze all, but the three last fine-tuned convolutional blocks.

Separately, thanks to R2’s suggestion, we transferred less sensitive models to texture to confirm further that shape bias is partially responsible for improving transferability.

---

### Decision · Program_Chairs · 2021-01-07
**Final Decision**

**Decision:**

Accept (Poster)

**Comment:**

The premise of the work is simple enough: investigate if networks that are trained with an adversarial objective end up being more suitable for transfer learning tasks, especially in the context of limited labeled data for the new domain. The work uncovers the fact that shape-biased representations are learned this way and this helps for the tasks they considered.

There was rather robust back and forth between the authors and the reviewers. The consensus is that this work has merit, has good quality experiments and investigates something with high potential impact (given the importance of transfer learning in general). I hope that most of the back and forth findings are incorporated in the final version of this work (especially the discussion and comparison with Shafahi et al., as well as all the nuances of the shape bias).